# A Multi-Fidelity Mixture-of-Expert Framework Integrating PDE Solvers and Neural Operators for Computational Fluid Dynamics

## Abstract

Solving Navier-Stokes equations is essential for computational fluid dynamics. While recent advancements in neural operators provide significant speed-ups, they often struggle to generalize to out-of-distribution scenarios. On the other hand, hybrid models that integrate neural networks with conventional numerical solvers offer improved generalization ability but incur high computational costs. To address this trade-off between computational efficiency and generalization ability, we propose the Multi-Fidelity Mixture-of-Experts (MF-MoE) framework. This framework combines a pure neural operator with multiple solver-based hybrid models of varying fidelity, leveraging them as expert models. A physics-aware gating network dynamically selects the most appropriate expert based on input characteristics, optimizing both computational cost and predictive accuracy. This innovative design enables faster inference for in-distribution inputs while ensuring better generalization for out-of-distribution cases. Extensive experiments on fluid flow prediction governed by the incompressible Navier-Stokes equations demonstrate that MF-MoE consistently outperforms baseline approaches, offering an efficient solution for PDE surrogate modeling.

## 1 Introduction

Computational fluid dynamics governed by nonlinear partial differential equations (PDEs) are ubiquitous in scientific and engineering applications, including the earth system modeling (Palmer & Stevens, 2019; Gelbrecht et al., 2023), fluid flow prediction (Belbute-Peres et al., 2020; Ma et al., 2024), gas leak detection (Lee et al., 2024), urban water clarification (Li & Shatarah, 2024; Putra et al., 2024), and blood flow modeling (Schwarz et al., 2023; Csala et al., 2024). Accurately resolving these PDEs at high spatial and temporal resolutions often remains computationally prohibitive, as conventional numerical methods demand immense computational resources and long runtimes.

Recently, neural operator methods have emerged as a promising alternative, offering data-driven PDE approximations that can exploit GPU acceleration for much faster training and inference (Li et al., 2020c; Wen et al., 2022; Janny et al., 2023; Azizzadenesheli et al., 2024; Navaneeth et al., 2024; Xiong et al., 2024; Raonic et al., 2024; Liang et al., 2024; Wu et al., 2024; Li et al., 2024). Despite these advantages, a pure neural network commonly struggles when confronted with out-of-distribution inputs, as demonstrated by Belbute-Peres et al. (2020) and Sun et al. (2023). This limitation hampers their robustness and restricts their applicability in real-world scenarios.

To address these shortcomings, hybrid approaches that integrate classical PDE solvers into neural architectures have been proposed (Mishra, 2018; Um et al., 2020; Belbute-Peres et al., 2020; Zhuang et al., 2021; Dresdner et al., 2022; Sun et al., 2023; Wang et al., 2024; Cao et al., 2024; Liu et al., 2024). Instead of directly learning a full-resolution PDE solution, these solver-embedded strategies employ a neural network to refine an inaccurate solution produced by a lower-resolution solver. For instance, Belbute-Peres et al. (2020) introduced a graph-based model that corrects the up-scaled coarse-mesh outputs from computational fluid dynamics simulations, reducing the out-of-distribution prediction error of a pure graph neural network model in the airfoil air flow prediction task.

However, while hybrid approaches improve the generalization ability compared to pure neural network models, they inevitably slow down training and inference due to the integration of numerical

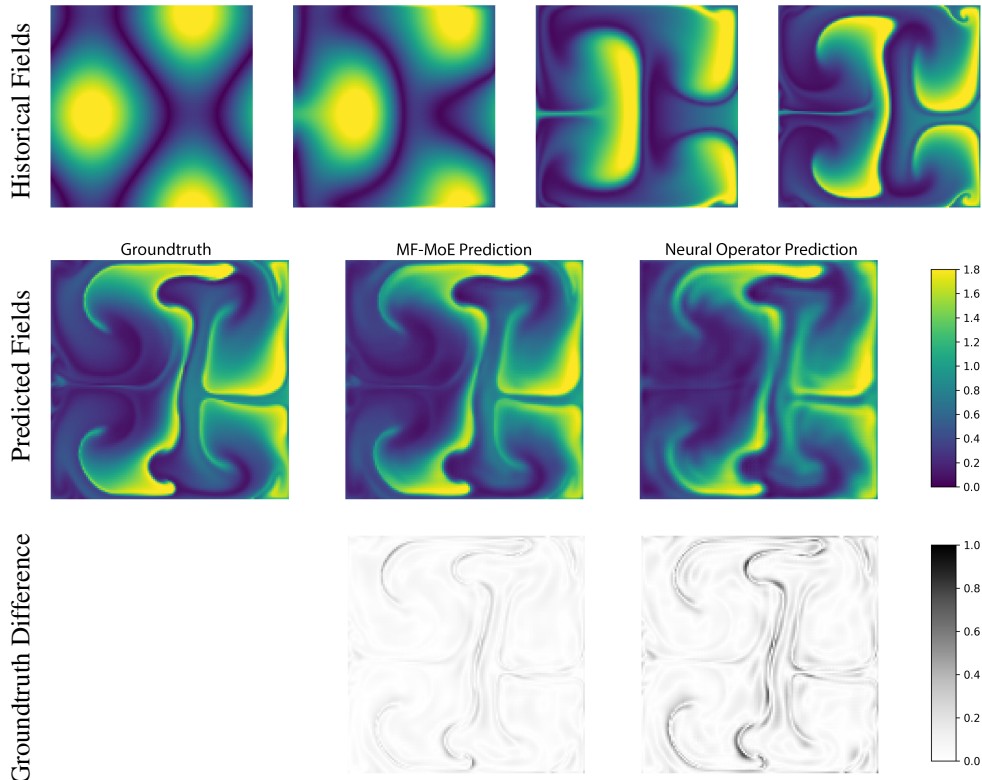

Figure 1: Comparison and visualization of fluid flow predictions from our proposed MF-MoE framework (with the base model: UNO-64) and the neural operator (UNO-64). The top row shows four frames of historical fields used as input for the models with the viscosity $\mu = 0.002$. The middle row compares ground-truth fields with the prediction from MF-MoE framework and the neural operator prediction. In the bottom row, each panel visualizes the absolute difference between the predictions and ground-truth fields (with darker regions indicating larger errors).

PDE solver[1]. As the result, one can rely on solver-based refinement for improved accuracy on out-of-distribution data, but must then accept the associated computational overhead. This conflict motivates the following natural question:

> **Q:** *Can we design a model architecture that enjoys fast inference time for in-distribution samples while maintaining high generalization ability for out-of-distribution samples?*

We present an affirmative answer to this question by introducing the **Multi-Fidelity Mixture of Experts (MF-MoE)**, a mixture-of-experts framework that seamlessly integrates multiple solver-based hybrid models of varying fidelity with a pure neural operator. As shown in Figure 1, the simulation result indicates that the MF-MoE predictions show markedly smaller error regions, highlighting the improved accuracy achieved by our proposed framework. This architecture dynamically balances computational efficiency and generalization ability by utilizing a physics-aware gating network to select the most suitable expert for each input.

Furthermore, we propose a constrained optimization framework for training MF-MoE, employing a Lagrangian relaxation strategy. Unlike traditional approaches that treat regularization terms as fixed

---

[1]In some specific architectures (e.g., the Frozen Mesh Mode of the CFD-GCN model by Belbute-Peres et al. (2020)), the PDE solution over the training sample can be separately cached. In such cases, training time is not significantly affected, but the inference cost remains unavoidable, which is also our main focus.

hyperparameters, this formulation explicitly incorporates time cost as a constraint. Extensive experiments on fluid flow prediction governed by the incompressible Navier-Stokes equations demonstrate that MF-MoE consistently achieves superior performance while maintaining acceptable inference time, validating the effectiveness and efficiency of the proposed MF-MoE framework.

## 2 RELATED WORK

**Physics-Informed Neural Network (PINN) and Neural Operator Learning**  Physics-Informed Neural Networks (PINNs) provide a direct approach to solving PDEs by parameterizing the solution as a neural network, which is trained to satisfy the governing equations and boundary conditions (E & Yu, 2018; Raissi et al., 2019; Bar & Sochen, 2019; Smith et al., 2020; Wang et al., 2022). However, as noted by Sun et al. (2023), PINNs require re-optimization for every new setup, limiting their scalability, particularly for dynamic problems. An alternative neural network-based method for solving PDEs is neural operator learning (Lu et al., 2019; Bhattacharya et al., 2020; Patel et al., 2021; Li et al., 2020c; Tran et al., 2021b; Gupta et al., 2021b; Nelsen & Stuart, 2021; Cao et al., 2021; Li et al., 2020b;a; Liang et al., 2024; Wu et al., 2024; Li et al., 2024). Unlike PINNs which parameterize the solution directly, neural operator methods learn a parameterized representation of a mapping from a field over the spatial-temporal domain (e.g., initial or boundary conditions) to another field over the same domain. These methods extend PDE solutions to functional mappings, allowing resolution-agnostic predictions through techniques such as Fourier transforms (Li et al., 2020c; Tran et al., 2021a) and wavelet transforms (Gupta et al., 2021a). However, as highlighted by Sun et al. (2023) and Belbute-Peres et al. (2020), purely neural network-based structures often struggle with out-of-distribution data due to the tendency of over-parameterized neural networks to overfit (Lawrence et al., 1997).

**Solver-Based Hybrid Models**  Recent advancements have introduced various neural network designs that integrate with classical numerical simulators (Holl & Thuerey, 2024; Thuerey et al., 2021; Economon et al., 2016; Anderson et al., 2021; mfem), broadly categorized into two approaches. The first approach employs neural networks to learn the stencils of advection-diffusion problems within the Finite Volume Method (FVM) framework (Bar-Sinai et al., 2019; Kochkov et al., 2021; Sun et al., 2023). The second approach leverages neural networks to correct numerical errors arising from low-resolution simulator outputs (Mishra, 2018; Um et al., 2020; Belbute-Peres et al., 2020; Pestourie et al., 2021; Dresdner et al., 2022; List et al., 2022; Frezat et al., 2022; Bruno et al., 2022; Ma et al., 2024). Our proposed MF-MoE framework falls into the latter category and aims to enhance time-efficiency by dynamically controlling the numerical solver's involvement.

## 3 BACKGROUNDS

In this section, we introduce the basic problem setting and highlight the observation that increasing spatial resolution will slow down the numerical PDE solver.

**Problem Setting**  In this work, we focus on the computational fluid dynamic problems characterized by the following form of *incompressible Navier-Stokes equations* (Temam, 1977):

$$\rho\left(\frac{\partial \mathbf{u}}{\partial t} + (\mathbf{u} \cdot \nabla)\mathbf{u}\right) = -\nabla \mathbf{p} + \mu \Delta \mathbf{u} + \mathbf{f}, \quad \text{with} \quad \nabla \cdot \mathbf{u} = 0, \tag{1}$$

where $\nabla$ is the vector differential operator, $\Delta := \nabla \cdot \nabla$ is the Laplace operator, $\mathbf{u}$ is the velocity field with the no-slip Dirichlet condition, $\mathbf{p}$ is the pressure, $\rho$ is the fluid density, $\mu$ is the constant viscosity, and $\mathbf{f}$ is an external force field. The incompressibility constraint is given by $\nabla \cdot \mathbf{u} = 0$. We consider a one-step flow prediction problem: given the physical fields from the previous $K$ time steps, $[\mathbf{u}_1, \ldots, \mathbf{u}_K]$, the objective is to predict the physical fields $\mathbf{u}_{K+1}$ at the next time step. This problem is widely studied in closed-box fluid dynamics, and many works adopt this setting (Gupta & Brandstetter, 2022; Ruhe et al., 2023; Lippe et al., 2023; Brandstetter et al., 2022).

**Numerical Solvers**  Solver-based hybrid models typically depend on external PDE solvers. In our work, we employ the PhiFlow framework (Holl & Thuerey, 2024), which has also been widely used in existing work (Um et al., 2020; Gupta & Brandstetter, 2022; Brandstetter et al., 2022; Schnell

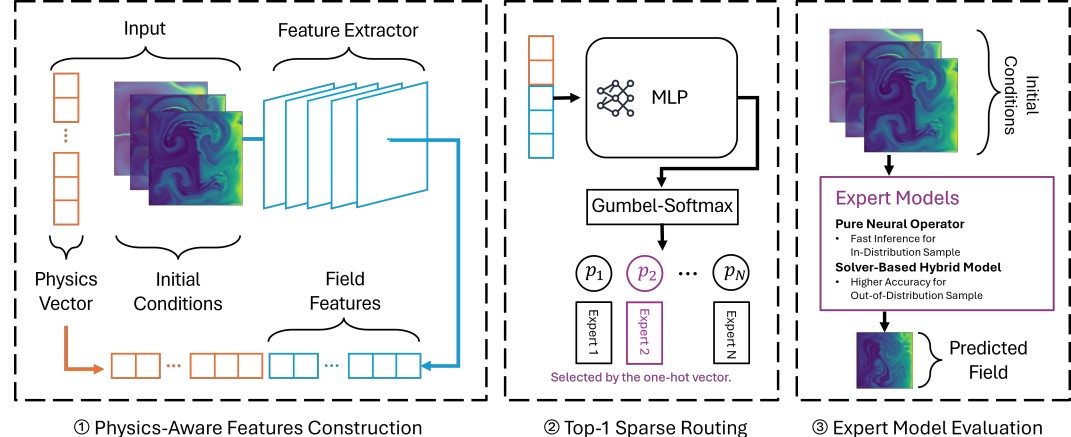

Figure 2: Overview of the forward pass of the MF-MoE framework: (1) Both the initial physical fields and the key physical parameters, such as the viscosity $\mu$ in Equation (1), are processed by the feature extractor to construct the Physics-Aware Feature. (2) The gating network produce the features from the previous step to a routing distribution. The Gumbel-Softmax ensures the Top-1 sparsely routing. (3) The selected expert (it can be a lightweight neural operator or a more expensive solver-based hybrid model) will further process the input fields to obtain the predicted field.

et al., 2022; Holl et al., 2022; Ruhe et al., 2023). This numerical solver employs the explicit forward Euler method to approximate the time evolution of the Navier-Stokes equations by discretizing the time derivative with a finite difference scheme. In general, if $\mathbf{u}^n$ denotes the velocity field at time $t_n$, then the forward Euler update from $t_n$ to $t_{n+1} = t_n + \Delta t$ is given by

$$\mathbf{u}^{n+1} = \mathbf{u}^n + \Delta t F\left(\mathbf{u}^n\right),$$

where $F(\mathbf{u}^n)$ represents the spatially discretized terms of the Navier-Stokes equations, including advection, diffusion, and external forcing. To enforce the incompressibility condition ($\nabla \cdot \mathbf{u}^{n+1} = 0$), a pressure projection step is performed. The method's accuracy is influenced by the time step size $\Delta t$ and the spatial resolution $(n_x, n_y)$. Larger $\Delta t$ can introduce larger numerical errors or even instability due to the Courant-Friedrichs-Lewy (CFL) condition (Courant et al., 1967), while coarser grids reduce spatial accuracy. We will see in Figure 4, finer grids improve accuracy but significantly increase computational costs due to the higher number of degrees of freedom.

## 4 THE MOE FRAMEWORK WITH MULTI-FIDELITY PDE SOLVERS

In this section, we present the structure of our proposed MF-MoE framework; its inference step is illustrated in Figure 2. The MF-MoE consists of two primary components:

- **Physics-Aware Gating Network**: This network learns to determine whether using a physical solver is necessary for a given input. If a solver is required, the gating network also selects the appropriate fidelity level for the hybrid models.

- **Multi-Fidelity Experts**: In candidate expert models, we consider one pure neural operator such as Fourier Neural Operators (FNOs) (Li et al., 2020c), or UNO (Ma et al., 2021; Chen & Thuerey, 2021; Rahman et al., 2022), and multiple solver-based hybrid models with varying fidelity levels. The pure neural operator offers significantly faster inference speeds but may have limited generalization capabilities. In contrast, the hybrid models enhance generalization by incorporating solver-based refinements but incur higher time costs.

### 4.1 PHYSICS-AWARE GATING NETWORK

The gating network helps for deciding which expert is most appropriate for a given input field. Unlike conventional mixture-of-experts models (in physics), of which routing processes focus solely on

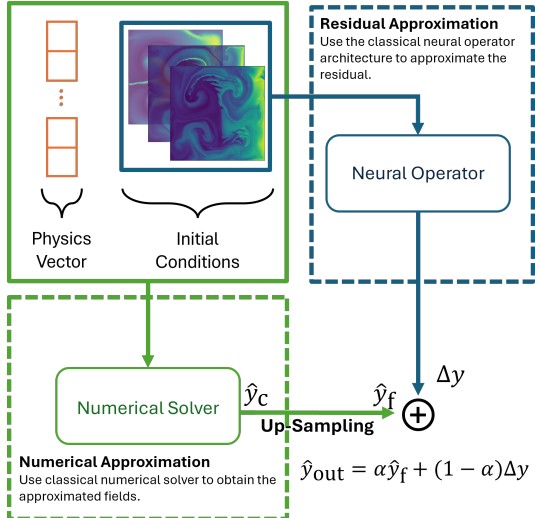

Figure 3: The illustration of the *solver-based hybrid model* used in this paper. The input sample is processed by the numerical solver and the pure neural network. The output of the numerical solver is the coarse prediction $\hat{y}_c$; it will be up-sampled to the fine prediction $\hat{y}_f$ with the desired dimension using the $k$-nearest neighbor methods. The output of the pure neural network is used to approximate the residual term $\Delta y$. The prediction $\hat{y}_{\text{out}}$ is a convex combination of two terms: $\hat{y}_{\text{out}} := \alpha \hat{y}_f + (1 - \alpha)\Delta y$, where $\alpha \in [0, 1]$ is a hyper-parameter for tuning.

initial physical fields (Sharma & Shankar, 2024; Liang et al., 2024; Hao et al., 2023), our gating network also considers the inherent physical parameters relevant to the PDE problem, such as the dynamic viscosity. To pass these information to the gating network, we apply the **physics-aware features construction**. As illustrated in Step 1 of Figure 2, the input sample $x_{\text{input}} = (v_{\text{phys.}}, \mathbf{u}_{\text{fields}})$ is separately spitted into the physical information (denoted by Physics Vector $v_{\text{phys.}}$) and the initial fields (denoted by Initial Conditions $\mathbf{u}_{\text{fields}}$). The initial fields are processed using a simple multilayer perceptron (MLP), denoted by $\text{NN}_{\text{MLP}}$, to extract the field feature and concatenate it with the physical information.

$$v_{\text{field feat.}} = \text{NN}_{\text{MLP}}(\mathbf{u}_{\text{fields}}), \quad \text{and} \quad v_{\text{phys. feat.}} = \begin{bmatrix} v_{\text{field feat.}} \\ v_{\text{phys.}} \end{bmatrix}.$$

The vector $v_{\text{phys. feat.}}$ will be used in the gating network to decide the routing strategy. This concatenation trick has been widely used in other fields, e.g. Guo et al. (2022); Xie et al. (2021) for image segmentation, to merge extracted features from multiple sources.

## 4.2 MULTI-FIDELITY EXPERT MODELS

Our MF-MoE system integrates two distinct types of experts: (1) *Pure Neural Operator*. This expert performs inference with no calls to a PDE solver. In our paper, we mainly use FNOs (Li et al., 2020c), UNO (Ronneberger et al., 2015), and ResNet (Stachenfeld et al., 2021) as the base model; they are commonly used as the baselines in existing literature and excel in fast inference and high accuracy over in-distribution samples. (2) *Multi-Fidelity Solver-Based Hybrid Models*. This type of experts fuses a numerical PDE solver with a refinement neural network. In our paper, we use the hybrid model illustrated by Figure 3. Specifically, a numerical solver generates a low-resolution approximation $\hat{y}_c$, which is then processed by an up-sampling step to recover fine-scale details. The neural operators are used to learn the correction $\Delta y$. A convex combination of $\hat{y}_C$ and $\Delta y$ is used as the final prediction.

Depending on resource availability, multiple solver-based models of varying fidelity (e.g., extremely coarse vs. moderately coarse mesh) can be included in the MoE to offer finer control over the accuracy-speed trade-off. The gating network thus selects lower or higher fidelity solver-based experts as needed.

**Up-Sampling** After obtaining the coarse output $\hat{y}_c \in \mathbb{R}^{n_{w,c} \times n_{h,c}}$, we generate the high-resolution prediction $\hat{y}_f \in \mathbb{R}^{n_{w,f} \times n_{h,f}}$ by applying the $k$-nearest neighbor ($k$-NN) interpolation for each time channel (Dasarathy, 1991; Shakhnarovich et al., 2008; Belbute-Peres et al., 2020). For each scaled pixel location $(i_f, j_f)$ in $\hat{y}_f$, we identify the $k$ nearest neighbors $\{(i_c^{(n)}, j_c^{(n)})\}_{n=1}^k$ of $(i_f, j_f)$ in $\hat{y}_c$ based on the given metric $d$. Then the high-resolution pixel value is computed as a weighted average of these neighbors:

$$\hat{y}_f(i_f, j_f) = \frac{1}{W} \sum_{n=1}^k w_n \cdot \hat{y}_c(i_c^{(n)}, j_c^{(n)}),$$

where $w_n = \frac{1}{d\left((i_f, j_f), (i_c^{(n)}, j_c^{(n)})\right)}$ and $W = \sum_{n=1}^k w_n$. In this paper, we always set $k = 4$. Appendix G further visualizes the performance of this up-sampling procedure on an example field.

### 4.3 TRAINING THE MF-MOE FRAMEWORK

In this section, we describe how to train our proposed MF-MoE framework. Unlike most existing MoE approaches that focus on improving the prediction accuracy of neural operators or other hybrid models, our primary goal is to control the time cost introduced by incorporating an external numerical solver during inference while maintaining the prediction accuracy. This goal is obviously more challenging than existing well-studied tasks as it requires balancing two competing goals—preserving the generalization ability of hybrid models while ensuring computational tractability. To achieve this, we formulate the problem as a constrained optimization task:

$$\min_\theta \quad \mathbb{E}_{(x,y)\sim\mathcal{D}}\mathbb{E}_{\xi\sim\mathsf{G}_{\theta_{\text{gate}}}(x)} \left\|\mathsf{E}_{\theta_{\text{expert},\xi}}(x) - y\right\|^2 \tag{2}$$

$$\text{subject to} \quad \mathbb{E}_{(x,y)\sim\mathcal{D}}\mathbb{E}_{\xi\sim\mathsf{G}_{\theta_{\text{gate}}}(x)}\mathcal{T}\left(\mathsf{E}_{\theta_{\text{expert},\xi}}\right) \leq c,$$

where $\theta = (\theta_{\text{gate}}, \theta_{\text{expert},1}, \ldots, \theta_{\text{expert},N})$ denote all trainable parameters of the MF-MoE framework with $N$ experts, $(x, y)$ is a data pair sampled from the data distribution $\mathcal{D}$, the mapping $\mathsf{G}_{\theta_{\text{gate}}} : x \mapsto [0, 1]^N$ is the gating network parameterized by the parameter $\theta_{\text{gate}}$ that produces a distribution over $N$ experts for each input $x$, the operator $\mathsf{E}_{\theta_{\text{expert},\xi}}$ is the $\xi$-th expert parameterized by $\theta_{\text{expert},\xi}$, $\mathcal{T}$ is the time-cost operator mapping a (hybrid) neural operator model to its inference time, and $c$ is the permissible time-cost threshold.

To solve this constrained optimization problem, we adopt a Lagrangian relaxation strategy (Beavis & Dobbs, 1990) that converts the constraint into a penalty term in the objective function. Concretely, let $\lambda \geq 0$ be a Lagrange multiplier associated with the time-cost constraint. We form the following Lagrangian:

$$\mathcal{L}(\theta, \lambda) = \mathbb{E}_{(x,y)\sim\mathcal{D}}\mathbb{E}_{\xi\sim\mathsf{G}_{\theta_{\text{gate}}}(x)} \left\|\mathsf{E}_{\theta_{\text{expert},\xi}}(x) - y\right\|^2 + \lambda\mathbb{E}_{(x,y)\sim\mathcal{D}}\left(\mathbb{E}_{\xi\sim\mathsf{G}_{\theta_{\text{gate}}}(x)}\mathcal{T}\left(\mathsf{E}_{\theta_{\text{expert},\xi}}\right) - c\right),$$

$$\tag{3}$$

where the second term serves as a soft penalty on the expected time cost.

**Time Cost Operator $\mathcal{T}$** Instead of measuring the time cost for each forward pass on the fly, we pre-calculate the solver's runtime at different resolutions in advance and maintain it as a look-up table. The table used in this paper are reported in Table 1. Since the up-sampling step and the neural network's inference time are negligible (less than $0.2$ seconds), we only record the solver's time cost. As the result, the soft penalty in Equation (3) can be rewritten as

$$\lambda\left(\mathbb{E}_{\xi\sim\mathsf{G}_{\theta_{\text{gate}}}(x)}\mathcal{T}\left(\mathsf{E}_{\theta_{\text{expert},\xi}}\right) - c\right) = \lambda\left(\mathsf{G}_{\theta_{\text{gate}}}(x)^\top\mathcal{T} - c\right),$$

where $\mathcal{T} \in \mathbb{R}^N$ is the look-up table; the $\xi$-th entry of this vector represents the time cost of $\xi$-th expert model.

**Training Hard Gating Network** We use the Gumbel-Softmax reparameterization trick (Jang et al., 2016; Maddison et al., 2016) in the gating network to produce a nearly discrete selection of experts while preserving differentiability during backpropagation. Concretely, the gate neural network $G_{\theta_{\text{gate}}}$ receives the input data $x$ and outputs the logits of each expert. These logits are then passed through

Table 1: Configuration details for different resolution levels. Each configuration specifies the number of timesteps (nt), spatial resolution (nx, ny), time cost per frame, and MSE of the numerical solution compared to the ground truth (GT). The ground truth configuration achieves the highest accuracy with the highest computational cost, while coarser resolutions progressively reduce time cost at the expense of higher error. The visualization of an example data pair is provided in Appendix G to better illustrate the difference among different resolution levels.

| Resolution Level | nt | (nx, ny) | Time Cost (s) | MSE Error |
|---|---|---|---|---|
| Groundtruth (GT) | 512 | $(256, 256)$ | $1.58 \times 10^2$ | $< 0.001$ |
| Fine | 64 | $(128, 128)$ | $0.38 \times 10^2$ | 0.065 |
| Medium | 32 | $(64, 64)$ | $0.20 \times 10^2$ | 0.104 |
| Coarse | 16 | $(32, 32)$ | $0.12 \times 10^2$ | 0.135 |
| XCoarse | 8 | $(16,16)$ | $0.08 \times 10^2$ | 0.151 |

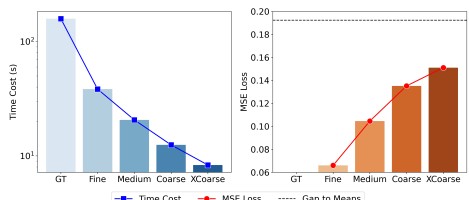

Figure 4: Computational cost (in seconds) versus spatial resolution in the numerical solver. Finer grids increase accuracy but require significantly more computational time. Details on the resolution configuration are provided in Appendix B.1. In the right panel, the black line labeled as "Gaps to Means", indicates the minimum MSE loss of using a constant field as the prediction.

a Gumbel Softmax layer with a fixed temperature $\tau = 0.1$ and the hard routing. Following the implementation of Paszke et al. (2019), we use the "Straight-Through" gradient estimator (Yin et al., 2019) so that, at inference, the routing is one-hot and selects exactly one expert, whereas during backpropagation the gradients flow through the soft routing probabilities. This design allows us to obtain a "hard" one-hot sample in the forward pass while still enabling gradients to update the gating probabilities $\mathsf{G}_{\theta_{\text{gate}}}(x)$ and thus the parameters $\theta_{\text{gate}}$ end-to-end alongside the expert parameters $\theta_{\text{expert},\xi}$. To further regularize the gating distribution and discourage trivial allocations (e.g., always selecting the same expert), we impose a KL divergence penalty on $\mathsf{G}_{\theta_{\text{gate}}}(x)$ toward a uniform prior $\mathsf{U}_N$ with weight $\beta_{\text{reg}} = 0.2$. We also decay $\beta_{\text{reg}}$ linearly over the course of training.

**Training Algorithm** Building on the time cost look-up table and the KL regularization, our objective reduces to minimizing the following Lagrangian function:

$$\widehat{\mathcal{L}}(\theta, \lambda) := \mathbb{E}_{(x,y)\sim\mathcal{D}}\mathbb{E}_{\xi\sim\mathsf{G}_{\theta_{\text{gate}}}(x)} + \beta_{\text{reg}}d_{\text{KL}}\big(\mathsf{G}_{\theta_{\text{gate}}}, \mathsf{U}_N\big) + \lambda\mathbb{E}_{(x,y)\sim\mathcal{D}}\Big(\mathsf{G}_{\theta_{\text{gate}}}(x)^\top\mathcal{T} - c\Big).$$

We solve this via the standard stochastic gradient descent-ascent procedure (Yan et al., 2020; Chen et al., 2022; Beznosikov et al., 2023). Specifically, we update $\theta$ with a stochastic gradient descent step; then update $\lambda$ with a one-step stochastic gradient ascent to ensure the feasibility of the time constraint.

$$\theta \leftarrow \theta - \eta_\theta \nabla_\theta \widehat{\mathcal{L}}(\theta, \lambda), \quad \text{and} \quad \lambda \leftarrow \max\Big\{0, \lambda + \eta_\lambda \big(\mathsf{G}_{\theta_{\text{gate}}}(x)^\top\mathcal{T} - c\big)\Big\},$$

where $\eta_\theta$ and $\eta_\lambda$ are the corresponding learning rates. In this paper, we replace the SGD optimizer used in the gradient descent step with the AdamW optimizer (Loshchilov et al., 2017; Loshchilov & Hutter, 2019) with setting the learning rate $\eta_\theta = 0.002$ for faster convergence while we maintain the SGD update for the $\lambda$ parameter with the learning rate $\eta_\lambda = 0.01$ with exponentially decaying at the rate $r = 0.999$.

## 5 EXPERIMENTS

### 5.1 FLUID FLOW DYNAMIC PREDICTION

In this section, we consider solving the fluid flow dynamic prediction problem given by the incompressible Navier-Stokes equation as defined in Equation (1). Given the input fields $[\mathbf{u}_t, \mathbf{u}_{t+1}, \mathbf{u}_{t+2}, \mathbf{u}_{t+3}]$, the objective is to predict the next time-step field $\mathbf{u}_{t+4}$. For this task, we adopt three foundational neural operator architectures: FNOs (Li et al., 2020c), UNOs (Ma et al., 2021; Chen & Thuerey, 2021; Rahman et al., 2022), and ResNet (Stachenfeld et al., 2021). These models are classical in the field of neural operators and are widely used as baselines in existing literature (Gupta & Brandstetter,

Table 2: Performance comparison on the fluid-flow prediction task described in Equation (1) across various neural operator architectures (including the FNO and UNO) and our proposed MF-MoE framework. The table reports the number of activated parameters, MSE metrics, and time constraint violation. The MF-MoE approach consistently achieves superior accuracy compared to the baseline models while significantly reducing solver-inference time compared to the numerical solvers.

| Ref. | Model[1] | Act. # Params | Train MSE | Valid MSE | Test MSE | $< 10$ secs?[2] |
|---|---|---|---|---|---|---|
| FNO (Li et al., 2020c) | FNO128-8$_{\text{modes16}}$ | 134 M | $2.71 \times 10^{-2}$ | $16.9 \times 10^{-2}$ | $17.0 \times 10^{-2}$ | |
| | FNO128-4$_{\text{modes16}}$ | 67.2 M | $5.43 \times 10^{-2}$ | $14.6 \times 10^{-2}$ | $15.1 \times 10^{-2}$ | ✓ |
| | FNO96-4$_{\text{modes32}}$ | 151 M | $6.22 \times 10^{-2}$ | $15.4 \times 10^{-2}$ | $15.7 \times 10^{-2}$ | |
| | FNO64-4$_{\text{modes32}}$ | 67.1 M | $9.33 \times 10^{-2}$ | $18.6 \times 10^{-2}$ | $17.5 \times 10^{-2}$ | |
| | MF-MoE (FNO64-4$_{\text{modes32}}$) | 67.1 M | $7.99 \times 10^{-2}$ | $13.4 \times 10^{-2}$ | $13.6 \times 10^{-2}$ | ✓ |
| UNO[3] (Ma et al., 2021) (Chen & Thuerey, 2021) (Rahman et al., 2022) | UNO-128 | 440 M | $0.13 \times 10^{-2}$ | $16.4 \times 10^{-2}$ | $16.4 \times 10^{-2}$ | |
| | UNO-64 | 110 M | $0.21 \times 10^{-2}$ | $17.7 \times 10^{-2}$ | $17.4 \times 10^{-2}$ | |
| | U-F3Net$_{\text{modes16,8,4}}$ | 187 M | $0.02 \times 10^{-2}$ | $11.7 \times 10^{-2}$ | $14.7 \times 10^{-2}$ | ✓ |
| | U-F2Net$_{\text{modes16,8}}$ | 175 M | $0.02 \times 10^{-2}$ | $11.5 \times 10^{-2}$ | $15.5 \times 10^{-2}$ | |
| | U-F1Net$_{\text{modes16}}$ | 160 M | $0.02 \times 10^{-2}$ | $13.2 \times 10^{-2}$ | $15.4 \times 10^{-2}$ | |
| | MF-MoE (UNO-64) | 110 M | $5.41 \times 10^{-2}$ | $10.8 \times 10^{-2}$ | $11.3 \times 10^{-2}$ | ✓ |
| ResNet[4] (Stachenfeld et al., 2021) | ResNet-128 | 1.2 M | $0.20 \times 10^{-2}$ | $6.96 \times 10^{-2}$ | $6.37 \times 10^{-2}$ | ✓ |
| | DilResNet-128 | 4.2 M | $0.80 \times 10^{-2}$ | $5.66 \times 10^{-2}$ | $5.40 \times 10^{-2}$ | |
| | MF-MoE (DilResNet-128) | 4.2 M | $0.36 \times 10^{-2}$ | $3.61 \times 10^{-2}$ | $5.14 \times 10^{-2}$ | ✓ |
| Numerical Solver (PhiFlow) | XCoarse ($16 \times 16$) | - | - | - | $15.1 \times 10^{-2}$ | ✓ |
| | Coarse ($32 \times 32$) | - | - | - | $13.5 \times 10^{-2}$ | ✗ |
| | Medium ($64 \times 64$) | - | - | - | $10.4 \times 10^{-2}$ | ✗ |
| | Fine ($128 \times 128$) | - | - | - | $6.50 \times 10^{-2}$ | ✗ |
| | Groundtruth ($256 \times 256$) | - | - | - | $< 0.001$ | ✗ |

[1] In both the FNO and UNO model, the first number indicate the number of hidden layers, which is used to control the size of the model. In the FNO model, the subscript 4$_{\text{modes32}}$ indicates that it retains 8 modes and uses 16 channels in its hidden layers. In the MF-MoE model, the bracket name indicates the choice of the base model, which is used as the pure neural operator expert in the MoE structure and the residual approximation model used in the hybrid structure (illustrated in Figure 3).
[2] We consider the averaged time cost over the test set and report if this averaged time violates the time constraint 10.0 seconds.
[3] We also consider the U-Nets with Fourier blocks belongs to a general category of U-shaped Neural Operator (UNO). Here the subscript represents the modes of Fourier blocks included in the architecture.
[4] ResNet (He et al., 2016) and Dilated (Atrous) Convolution (Chen et al., 2014; Yu & Koltun, 2016; Yu et al., 2017) were first used in computational vision tasks; Stachenfeld et al. (2021) applies these techniques in the fluid flow simulation. Here we use the same structure as (Stachenfeld et al., 2021). The number 128 indicates the depth of the network.

2022; Brandstetter et al., 2022; Ruhe et al., 2023). While numerous state-of-the-art models have been developed in recent years, most of them are based on these three models. Therefore, we focus on these three for their broad representativeness. Additional discussions of other advancing models are included in Appendix A.

**Datasets** We create a customized dataset by splitting the train set, validation set, and test set via the conditional parameters of viscosity $\mu$. The split is described in Equation (4). The detailed dataset configuration and a sample trajectory visualization of each resolution setting are put in Appendix B.1 and Appendix G.

$$
\begin{aligned}
\text{(Train)} \quad & \mu \in \{3.2, 1.6, 0.8\} \times 10^{-2}, \\
\text{(Valid)} \quad & \mu \in \{1.6, 0.8, 0.4, 0.2\} \times 10^{-2}, \\
\text{(Test)} \quad & \mu \in \{0.8, 0.4, 0.2, 0.1\} \times 10^{-2}.
\end{aligned}
\tag{4}
$$

**Training** We use the Mean Squared Error (MSE) loss on the predicted velocities as the evaluation metric. For the MF-MoE model, the number of experts $N = 4$, where one expert is a pure neural-operator model and the other three are solver-based hybrid models of varying fidelity (Fine, Medium, and Coarse); the time-cost constraint to $c = 10.0$, ensuring that a single solver-based hybrid model alone (though accurate) is excluded, as it exceeds this cost threshold. An additional discussion on the impact of the parameter $c$ is included in Section 5.2. All models are trained using the AdamW optimizer (Loshchilov et al., 2017; Loshchilov & Hutter, 2019) with a learning rate $\eta = 2 \times 10^{-4}$ for 8,000 training steps, a weight decay of $1 \times 10^{-5}$, and a batch size of 32. No further hyperparameter tuning is applied across different models. During the training of the MF-MoE framework, we update the parameter $\lambda$ with a naive stochastic gradient descent (SGD) step using an initial learning rate $\eta_\lambda = 0.01$. This learning rate is decayed exponentially with a factor $r = 0.999$. For the MF-MoE model, we introduce several additional hyperparameters: the number of experts $N = 4$, where

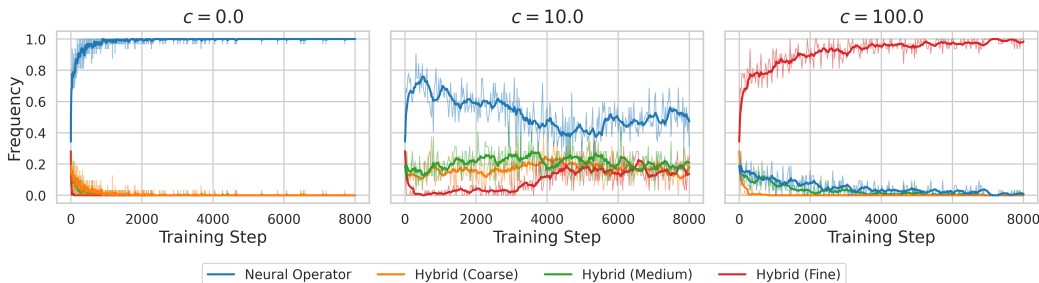

Figure 5: Illustration of the impact of the time constraint $c$ on the gating mechanism. As $c$ increasing, the gate network tends to choose the solver-based hybrid model with the highest fidelity.

one expert is a pure neural-operator model and the other three are solver-based hybrid models of varying fidelity (Fine, Medium, and Coarse). We set the time-cost constraint to $c = 10.0$, ensuring that a single solver-based hybrid model alone (though accurate) is excluded, as it exceeds this cost threshold.

**Experimental Results** In Table 2, we summarize the performance of our MF-MoE framework compared to baseline models. Here we present two primary observations: (1) **Improved Accuracy over Neural Operators.** The MF-MoE framework consistently outperforms neural operator baselines (FNO, UNO, and ResNet) across all evaluated scenarios. When using the same base architecture, MF-MoE achieves significant accuracy improvements over standalone neural operators. For instance, as visualized in Figure 1, MF-MoE with the UNO-64 base model exhibits markedly lower prediction error compared to the standalone UNO-64 neural operator, underscoring its enhanced modeling capability. (2) **Efficient Solver Integration.** In contrast to numerical solvers, which fail to meet the time constraint at every fidelity level except the XCoarse resolution level, the MF-MoE model satisfies the required time constraint $c = 10.0$. This indicates that MF-MoE greatly enhances the efficiency of solver-based hybrid models. In traditional designs, the inference time cost of such hybrid models is dominated by the numerical solver; however, MF-MoE circumvents this bottleneck by dynamically choosing the engagement of the numerical solver.

## 5.2 GATING MECHANISM

In this section, we examine the impact of the hyperparameter $c$ on the MF-MoE model. This parameter is crucial for balancing computational efficiency and prediction accuracy. In extreme cases, if $c$ is set to $+\infty$, the time constraint becomes inactive; consequently, the gating network will always select the most accurate expert, regardless of computational cost. To investigate this behavior systematically, we evaluate the model with $c$ values of 0.0, 10.0, and 100.0. As illustrated in Figure 5, when the time constraint is always active (i.e., $c = 0$), the gate selection degenerates to constantly choosing the pure neural operator, as the solver-based hybrid model incurs substantially higher penalties. Conversely, when the time constraint is consistently inactive (i.e., $c = 100$), the gate selection defaults to the solver-based model that yields the highest accuracy.

## 5.3 EXTENDED EXPERIMENTS: OTHER OUT-OF-DISTRIBUTION PARAMETERS

To further evaluate the generalization capabilities of the MF-MoE framework, we extend our assessment beyond the viscosity parameter $\mu$. As highlighted in Equation (1), the external force field $\mathbf{f}$ is a governing term driving the fluid dynamics, which is partially controlled by the buoyancy. Therefore, we modified the data generation configuration to introduce a varying buoyancy.

We constructed a dataset split based on the buoyancy along the $y$-axis to create a distinct out-of-distribution scenario with each parameter 100 independent sampled data pairs:

$$\text{(Train \& Valid)} \quad \mathbf{f} \in \{0.3, 0.4, 0.5\}, \qquad \text{(Test)} \quad \mathbf{f} \in \{0.7\}.$$

This setup tests the model's ability to extrapolate to highly perturbed flows that exhibit more chaotic turbulence than the training set. We compared the MF-MoE (using UNO-64 as the base) against a

Table 3: Performance comparison on out-of-distribution buoyancy force ($\mathbf{f} \in \{0.7\}$). The MF-MoE maintains lower error compared to the pure UNO model.

| Model | Test MSE | Valid Constraint |
|---|---|---|
| UNO-64 | $19.4 \times 10^{-2}$ | ✓ |
| MF-MoE (UNO-64) | $12.5 \times 10^{-2}$ | ✓ |
| Numerical Solver (Fine) | $6.44 \times 10^{-2}$ | ✗ |

Table 4: Comparison of multi-step prediction performance. We report the time-averaged MSE at different rollout horizons ($T$).

| Model | Time-Averaged MSE ($\times 10^{-2}$) | | |
|---|---|---|---|
| | $T = 5$ | $T = 10$ | $T = 20$ |
| FNO-128 | 21.5 | 24.8 | 28.1 |
| UNO-64 | 16.2 | 20.5 | 23.4 |
| ResNet-128 | 19.1 | 22.8 | 25.6 |
| MF-MoE (ResNet-128) | **9.45** | **12.3** | **15.1** |

standalone UNO-64 model. The results are summarized in Table 3. As hypothesized, the pure neural operator struggles to generalize, resulting in a high MSE of $19.4 \times 10^{-1}$. The operator tends to under-predict the velocity magnitudes, while the MF-MoE can be corrected by the numerical solver.

## 5.4 EXTENDED EXPERIMENTS: MULTI-STEP PREDICTIONS

In our main experimental results, we restricted our analysis to one-step predictions. In this subsection, we extend our evaluation to multi-step, long-horizon forecasting using an autoregressive rollout strategy, where predicted fields $\hat{u}_{t+1}$ serve as inputs for predicting $\hat{u}_{t+2}$. We compare the time-averaged MSE over a trajectory of $T$ time steps against baseline neural operators. As summarized in Table 4, the MF-MoE framework significantly mitigates the drift typically observed in pure neural operators, confirming its advances in the multi-step long-horizon setting.

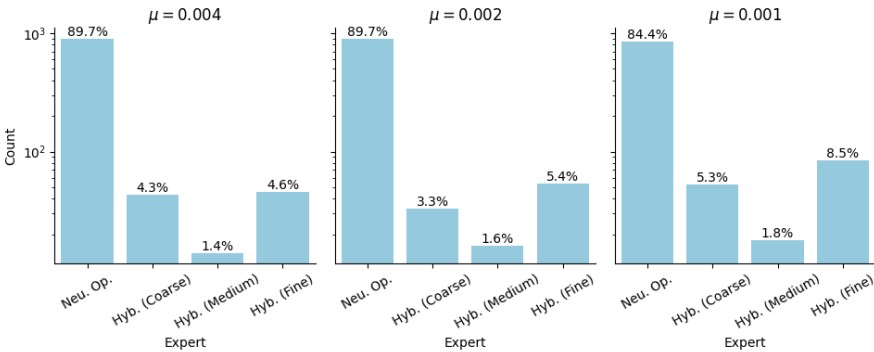

Figure 6: The histograms illustrate the frequency with which the gating network selects each expert on the selected test set. As the fluid viscosity decreases (indicating more out-of-distribution), the model increasingly routes to the high-fidelity expert (rising from 4.6% to 8.5%) while reducing routing to the pure neural network (dropping from 89.7% to 84.4%), demonstrating the framework's adaptive trade-off between efficiency and accuracy.

## 5.5 EXTENDED VISUALIZATION: DISTRIBUTION OF EXPERT SELECTIONS

In this subsection, we examine the distribution of expert selections made by the gating network during the inference phase, specifically investigating how the routing strategy correlates with the physical complexity of the input fluid dynamics. To validate this, we visualize the frequency of expert selection across test samples with varying viscosities $\mu \in \{0.004, 0.002, 0.001\}$. As illustrated in Figure 6, we observe a distinct shift in the routing distribution, confirming that the MF-MoE framework successfully learns to balance the trade-off between computational efficiency and generalization ability, autonomously identifying when the higher cost of a numerical solver is justified by the need for error correction in out-of-distribution scenarios.

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

# A  NEURAL OPERATORS

Recent advancements in neural operators have expanded beyond foundational architectures like FNOs (Li et al., 2020c) and U-Nets (Ronneberger et al., 2015; Ma et al., 2021; Chen & Thuerey, 2021; Rahman et al., 2022). Below, we summarize key developments in the field.

## A.1  FOURIER NEURAL OPERATORS (FNOS)

The FNO architecture utilizes Fast Fourier Transforms (FFTs) to decompose inputs into low and high Fourier modes, capturing global and local information essential for modeling physical fields. Beyond classical 2D/3D applications, recent efforts focus on improving scalability, stability, and accuracy for highly non-linear PDEs. For instance, domain-decomposition-based FNOs (Nidhan et al., 2024) and M2M (Liang et al., 2024) split large computational domains into subdomains to handle extreme-scale problems in fluid and solid mechanics. Multi-resolution variants, such as MFNO (Zhang et al., 2024), combine FNO blocks at different frequency levels to capture multi-scale phenomena, especially in turbulent flows. Notable extensions include U-FNOs (Wen et al., 2022) and IFNOs (You et al., 2022), which enhance the classical FNOs by extending their application from 2D to 3D predictions. Recent work by Rao et al. (2025) demonstrates FNO's effectiveness in solving turbulent Rayleigh-Bénard convection, outperforming traditional surrogates like Dynamic Mode Decomposition while achieving zero-shot super-resolution capabilities.

## A.2  U-SHAPED NEURAL OPERATORS (UNOS)

U-Net-based neural operators for PDE surrogates have seen continuous innovations aimed at capturing complex, multi-scale structures in physical fields. When replacing the convolution blocks in both sampling paths with the Fourier blocks, the U-Net becomes the U-shaped Neural Operator (UNO). Alongside wide Residual Networks (Zagoruyko, 2016) and spatial attention mechanisms (Vaswani, 2017), the Point Cloud Neural Operator (PCNO) (Huang et al., 2025) extends U-Net principles to handle parametric PDEs on complex geometries using adaptive meshing and rotation-invariant inputs. Recent innovations also include physics-informed latent architectures like PI-Latent-NO (Karumuri et al., 2025), which combines reduced-order modeling with operator learning for efficient training on high-dimensional PDEs. These developments underscore U-Nets' flexible design, enabling more accurate and efficient solution operators for a range of partial differential equations.

## A.3  GRAPH NEURAL OPERATORS

Graph Neural Operators (GNOs) extend the neural operator framework to irregular and graph-structured domains, enabling function space mappings on unstructured grids and complex geometries. By leveraging graph neural networks (GNNs), these architectures effectively model physical systems with non-Euclidean structures. The Spatio-Spectral Graph Neural Operator (Sp2GNO) (Sarkar & Chakraborty, 2024) integrates spatial and spectral GNNs to learn solution operators across arbitrary geometries. By combining local and global feature extraction, Sp2GNO can handle both time-dependent and time-independent PDEs on irregular computational domains. The Multipole Graph Neural Operator (MGNO) (Li et al., 2020a) addresses scalability challenges in PDE modeling by employing multipole expansions to capture long-range dependencies efficiently. The Physics-Informed Graph Neural Operator (PIGNO) (Chen et al., 2024) incorporates physical constraints into the operator learning process, ensuring solutions adhere to underlying laws governing the modeled system. Recent developments in GNOs highlight their versatility in handling PDEs on irregular domains, from computational mechanics to networked dynamical systems, further expanding the applicability of neural operators beyond structured grids.

## A.4  OTHER ADVANCING NEURAL OPERATOR ARCHITECTURES

Beyond FNO and U-Net variants, several novel frameworks have emerged. The Neural Operator-based symbolic Model approximaTion and discOvery (NOMTO) (Li et al., 2025) enables discovery of differential equations with singularities and special functions, significantly expanding symbolic regression capabilities. Hydraulic-based graph neural networks like mSWE-GNN (Bentivoglio et al., 2025) introduce multi-resolution modeling with ghost cells for flood prediction, achieving

700x speedups while handling time-varying boundary conditions. Additional architectures include Residual Networks for function space mapping (Cao et al., 2023), Wavelet Neural Operators (WNO) (Navaneeth et al., 2024), Koopman Neural Operators (Xiong et al., 2024), the U-shaped Neural Operator (Azizzadenesheli et al., 2024), and Implicit Adaptive Fourier Neural Operators (IAFNO) (Jiang et al., 2025), demonstrating neural operators' growing versatility in handling diverse physical systems, from turbulent flows to symbolic equation discovery.

## B GENERAL EXPERIMENTAL SETTINGS

In this appendix, we provide a detailed description of the experimental setup, including hardware and software specifications, data generation configurations, and other training details. Since the solver-based hybrid model (and the MF-MoE models that include these hybrid models as experts) can require days for training and testing, we pre-generate the solver simulation results of the provided dataset for faster processing. The full codes and other omitted experimental settings are included in the supplymentary material.

**Hardware Specifications**    All experiments were run on a single compute node with: Dual AMD EPYC 9124 CPUs (32 total CPU cores), 768 GB of DDR5 4800 MHz memory, and 8 NVIDIA RTX 6000 Ada Generation GPUs.

**Software Requirements**    Throughout all experiments, we use Python 3.11.2, CUDA 12.4.1, and OpenSSL 1.1.1k FIPS (25 Mar 2021). All other required Python packages match those specified by the original repository, PDEArena (Gupta & Brandstetter, 2022; Brandstetter et al., 2022; Ruhe et al., 2023).

### B.1 CONFIGURATION FOR DATA GENERATION

We adapt the existing data generation configuration from PDEArena's Github Repository, which has been widely used by Gupta & Brandstetter (2022); Brandstetter et al. (2022); Ruhe et al. (2023). More explicitly, we made the following modifications on the original configuration `navierstokes2dsmoke.yaml` to improve the accuracy of the PhiFlow's numerical solution: We reduced `skip_nt`, the initial timesteps to ignore, from 8 to 0, and reduced the `tmax`, the maximum time period, from 108.0 to 60.0; as the result, we are investigating the fluid dynamic over the time period $[18.0, 60.0]$. We control the overal numerical accuracy by adjusting the number of timesteps and the spatial resolution. More explicitly, these configurations are summarized in Table 5. The time cost and the MSE error is evaluated per frame. For example, if a trajectory with setting `nt` $= 64$ and `sampling_rate` $= 8$, it outputs $\frac{\texttt{nt}}{\texttt{sampling\_rate}} = \frac{64}{8} = 8$ frames; then its time cost is the total time cost $t_{\text{total}}$ divided by 8. Similarly, we take the slice of the same time step $t$, and evaluate the MSE loss between the up-sampled low-resolution field and the groudtruth field.

Table 5: Configuration details for data generation at different resolution levels. Each configuration specifies the number of timesteps (`nt`), spatial resolution (`nx`, `ny`), time cost per frame, and MSE of the numerical solution compared to the ground truth (GT). The ground truth configuration achieves the highest accuracy with the highest computational cost, while coarser resolutions progressively reduce time cost at the expense of higher error. The visual illustration has been presented in Figure 4.

| Resolution Level | Configuration File | nt | (nx,ny) | Time Cost (s) | MSE Error |
|---|---|---|---|---|---|
| Groundtruth (GT) | navierstokes2dsmoke_high_res.yaml | 512 | (256, 256) | $1.58 \times 10^2$ | $< 0.001$ |
| Fine | navierstokes2dsmoke_fine.yaml | 64 | (128, 128) | $0.38 \times 10^2$ | 0.065 |
| Medium | navierstokes2dsmoke_medium.yaml | 32 | (64, 64) | $0.20 \times 10^2$ | 0.104 |
| Coarse | navierstokes2dsmoke_coarse.yaml | 16 | (32, 32) | $0.12 \times 10^2$ | 0.135 |
| XCoarse | navierstokes2dsmoke_xcoarse.yaml | 8 | (16,16) | $0.08 \times 10^2$ | 0.151 |

For each viscosity $\mu$ (as shown in Equation (4) or the following dataset split), we generate 50 independent trajectories with the length 8 (they contain 250 pairs of data for each $\mu$ in total). As the result, the train, validation, and test set contain 750, 1000, and 1000 pairs of data, respectively. (for the case `nt` larger than 8, we set the `sampling_rate` parameter to omit the intermediate data). In Appendix G, we visualize a single trajectory for better illustrating the accuracy of the numerical

solution over different fidelity.

$$\text{(Train)} \quad \mu \in \left\{ 3.2, 1.6, 0.8 \right\} \times 10^{-2},$$

$$\text{(Valid)} \quad \mu \in \left\{ 1.6, 0.8, 0.4, 0.2 \right\} \times 10^{-2},$$

$$\text{(Test)} \quad \mu \in \left\{ 0.8, 0.4, 0.2, 0.1 \right\} \times 10^{-2}.$$

Other parameters are included in the supplementary material. We omit the full discussions on these parameters and the details of introduction can be found in the PDEArena (Gupta & Brandstetter, 2022).

## B.2 CONFIGURATION FOR TRAINING

In this section, we present the details on the training configuration. We use the AdamW optimizer (Loshchilov et al., 2017; Loshchilov & Hutter, 2019) with learning rate $\eta = 2 \times 10^{-4}$ for 8,000 training steps, a weight decay of $1 \times 10^{-5}$, and a batch size of 32. No further hyperparameter tuning is applied across different models. During the training of the MF-MoE framework, we update the parameter $\lambda$ with a naive stochastic gradient descent (SGD) step using an initial learning rate $\eta_\lambda = 0.01$. This learning rate is decayed exponentially with a factor r = 0.999, leading to a final effective rate of around 0.001. For the MF-MoE model, we introduce several additional hyperparameters: the number of experts $N = 4$, where one expert is a pure neural-operator model and the other three are solver-based hybrid models of varying fidelity (Fine, Medium, and Coarse). Their relative accuracy and time costs appear in Table 5. We set the time-cost constraint to $c = 10.0$, ensuring that a single solver-based hybrid model alone (though accurate) is excluded, as it exceeds this cost threshold. A concise summary of all the important hyperparameters is provided in Table 6.

Table 6: Hyperparameter overview for the training process in the fluid flow dynamic prediction experiment

| Hyperparameter | Value |
|---|---|
| Optimizer | AdamW |
| Learning rate ($\eta_\theta$) | $2 \times 10^{-4}$ |
| Number of training steps | 8,000 |
| Weight decay | $1 \times 10^{-5}$ |
| Batch size | 32 |
| $\lambda$ update | Naive SGD |
| $\lambda$ learning rate ($\eta_\lambda$) | 0.01 |
| Exponential decay rate (r) | 0.999 |
| Number of experts ($N$) | 4 |
| Time-cost constraint ($c$) | 10.0 |

## B.3 CONFIGURATION OF BASELINES

We list here a brief overview of the three types of model we have considered in this work. Their implementations are directly taken from the PDEArena (Gupta & Brandstetter, 2022) and we follow the same name used in their codes. These models are also compatible with our MF-MoE framework and can be served as the pure neural operator expert or the residual approximation network used in the solver-based hybrid expert.

- **Fourier Neural Operators (FNOs)** (Li et al., 2020c): FNOs are among the most widely used neural PDE surrogates, leveraging the Fast Fourier Transform (FFT) (Van Loan, 1992; Cooley & Tukey, 1965). In this framework, low-frequency Fourier modes capture global features, while high-frequency modes capture local details. The first set of parameters (e.g., 128, 96, 64) specifies the number of hidden channels, which determines the model size. The second parameter indicates the number of modes and the hidden channels of each mode.
- **ResNet** (He et al., 2016): The implementation of ResNet from (Gupta & Brandstetter, 2022) is different from the original implementation; we refer the reader for more details in the PDEArena (Gupta & Brandstetter, 2022). The number (256 and 128) is the number of hidden channels, which

is used to control the model size and the prefix "Dil" indicates if the Dilated ResNets (Stachenfeld et al., 2021) are used or not.

- **U-Net** (Ma et al., 2021; Chen & Thuerey, 2021; Rahman et al., 2022): The U-Net architecture, originally developed by Ronneberger et al. (2015) for biomedical image segmentation, has since been adapted for PDE surrogates (Ma et al., 2021; Chen & Thuerey, 2021). A special U-Net architecture with replacing all blocks with the Fourier blocks is called UNO, which is developed by Rahman et al. (2022). We adjust the number of hidden layers from 64 to 128 to control the size of UNO. If we only replace the blocks in downsampling paths, the model is called U-Nets with Fourier blocks. We consider different variants with including different modes in the Fourier blocks in our experiment.

## C  IMPACT STATEMENT

This paper presents a framework for accelerating the solver-based hybrid PDE surrogate model via the mixture-of-expert structure. Potential positive applications include more efficient climate simulations, fluid modeling, and biomedical analyses. We do not see any immediate negative societal impacts or ethical concerns specific to this approach.

## D  LIMITATIONS & FUTURE WORK

Despite the promising results demonstrated by the MF-MoE framework, several limitations remain, highlighting opportunities for future research: (1) This study primarily focuses on single-step predictions, which limits its applicability to scenarios requiring multi-step or long-horizon forecasts. Future work could explore incorporating unrolled training techniques and developing time-evolving gating strategies to enhance performance in these more complex settings. (2) While the MF-MoE framework is designed to be flexible and compatible with advanced models, its empirical performance with such models has not yet been thoroughly evaluated. Investigating its integration with state-of-the-art neural operators and hybrid models is a promising direction for future studies.

## E  THE USE OF LARGE LANGUAGE MODELS (LLMS)

We use LLMs to assist the writing refinement, the main article drafting, and the experimental codes generation. All generated details are carefully checked by experienced human.

## F  CONCLUSION

In this work, we introduced the **Multi-Fidelity Mixture-of-Experts (MF-MoE)** framework, a novel approach to operator learning that seamlessly integrates pure neural operators with the numerical solver. Leveraging a physics-aware gating network to dynamically route inputs to the most suitable expert, the framework achieves an optimal balance between computational efficiency and generalization ability by adopting the time cost as a constraint. It enables fast inference for in-distribution data while ensuring generalization ability on out-of-distribution data. This was validated through the fluid flow prediction experiment, where the MF-MoE framework consistently outperformed its base model counterparts while maintaining the time cost constraint. We believe this work provides an efficient approach for combining numerical solver and pure neural operator hence opens a new avenue for the development of solver-based hybrid models.

# G   VISUALIZATION OF NUMERICAL SOLUTION OVER DIFFERENT RESOLUTIONS

In this section, we present the visualization of numerical solution over different resolution levels (including Fine, Medium, Coarse, and XCoarse). Their detailed resolution is provided in Table 5.

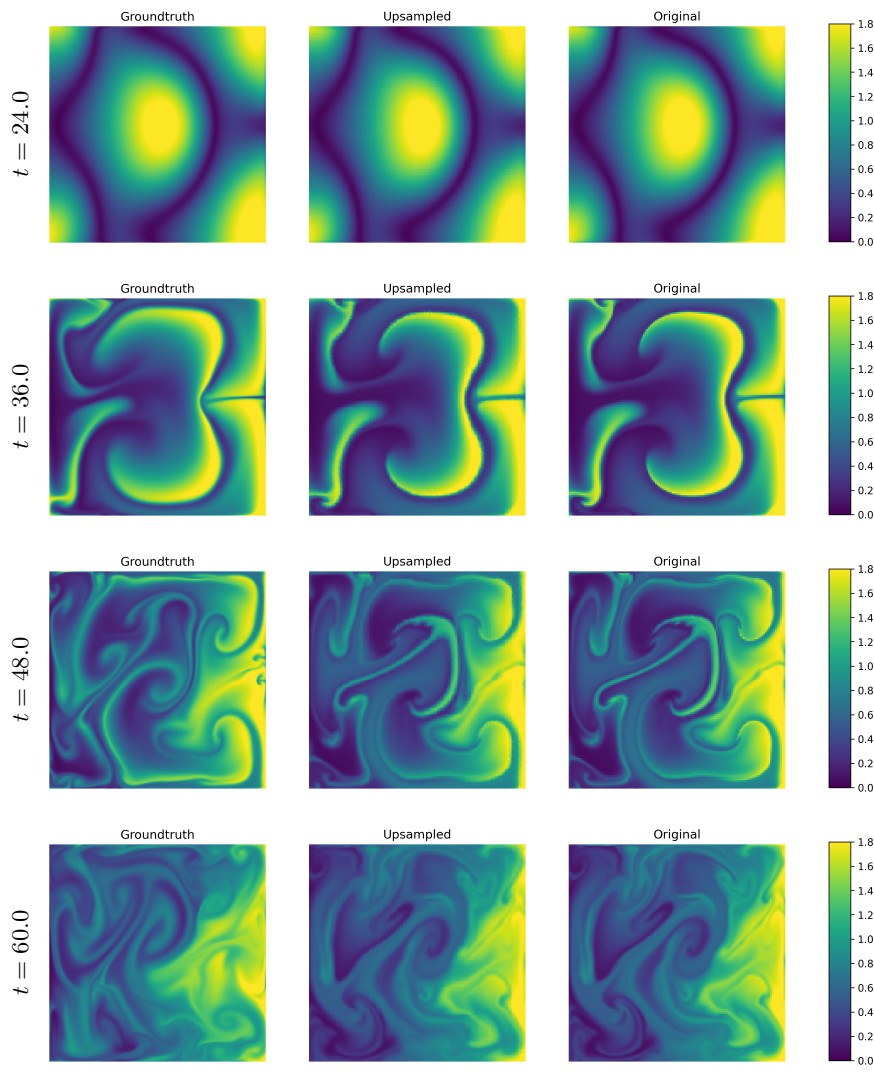

Figure 7: Illustration of the scalar field solved over the fine-level resolution at four time steps: $t = 24.0, 36.0, 48.0, 60.0$.

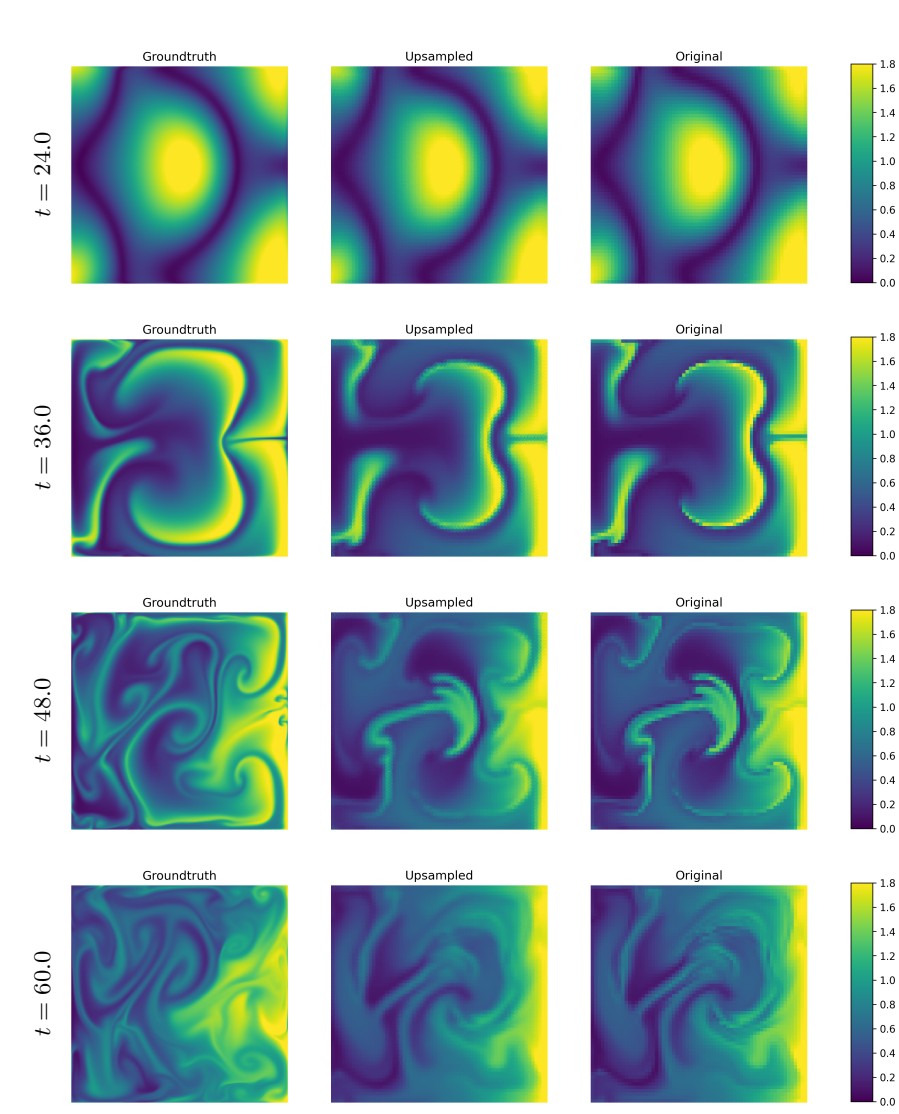

Figure 8: Illustration of the scalar field solved over the medium-level resolution at four time steps: $t = 24.0, 36.0, 48.0, 60.0$.

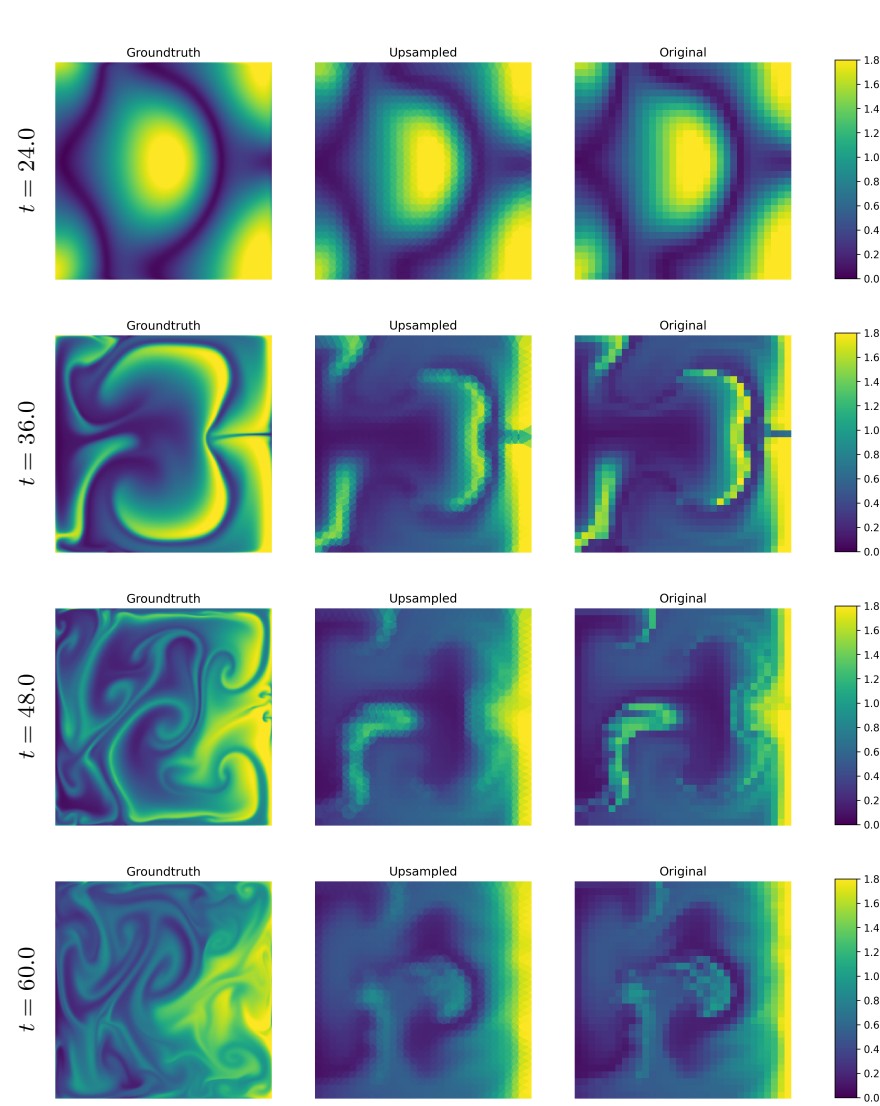

Figure 9: Illustration of the scalar field solved over the coarse-level resolution at four time steps: $t = 24.0, 36.0, 48.0, 60.0$.

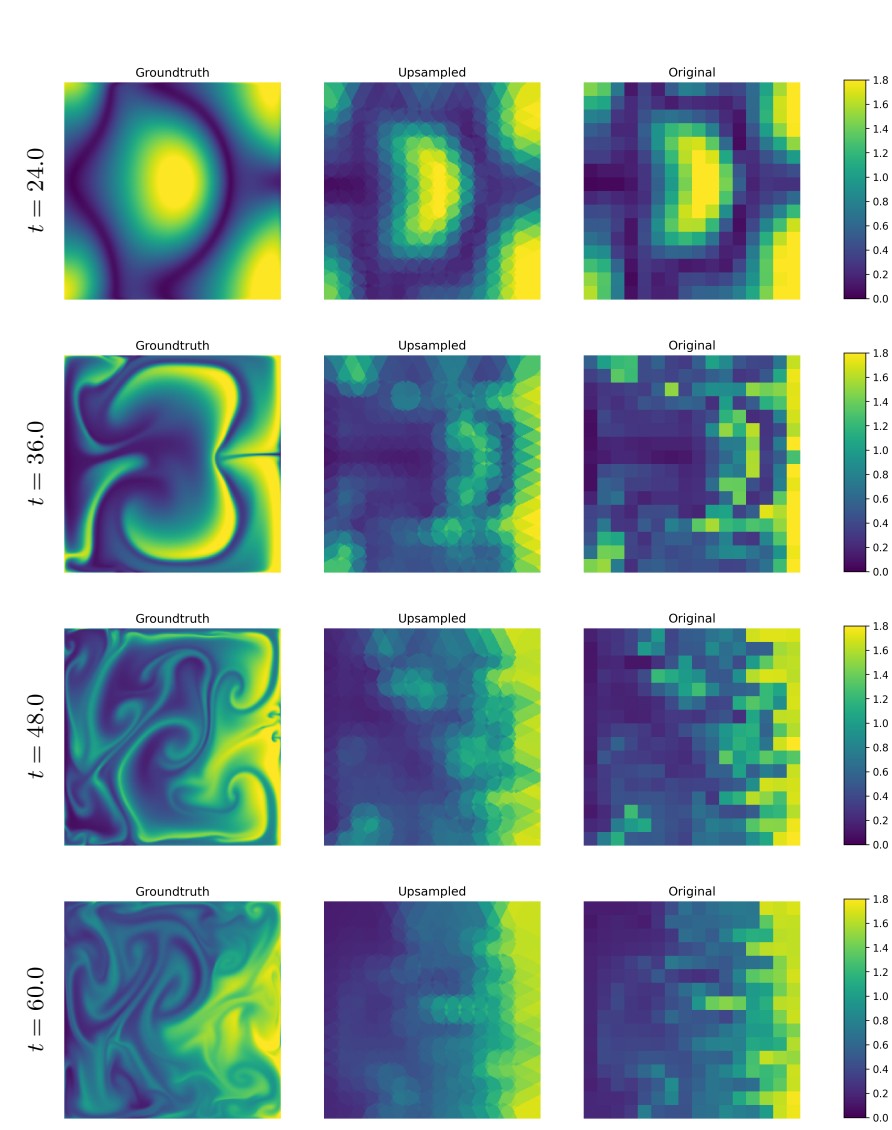

Figure 10: Illustration of the scalar field solved over the xcoarse-level resolution at four time steps: $t = 24.0, 36.0, 48.0, 60.0$.

