# OpenReview forum: "A Multi-Fidelity Mixture-of-Expert Framework Integrating PDE Solvers and Neural Operators for Computational Fluid Dynamics"
_ICLR.cc/2026/Conference — ICLR 2026 Conference Desk Rejected Submission_

### Official Review · Reviewer_sosU · 2025-10-26

**Soundness:** 2
**Presentation:** 2
**Contribution:** 2
**Rating:** 4
**Confidence:** 5

**Summary:**

This paper proposes a Multi-Fidelity Mixture-of-Experts (MF-MoE) framework that combines a pure neural operator with several solver-based hybrid models of varying fidelity. A physics-aware gating network is used to dynamically select the most suitable expert for each input, aiming to balance computational cost and generalization ability. The method is evaluated on incompressible Navier-Stokes fluid flow prediction tasks and claims to outperform baseline neural operators while respecting a time-cost constraint.

**Strengths:**

- The idea of combining multi-fidelity solvers with a gating mechanism is intuitive and relevant for CFD applications.

- The use of a constrained optimization formulation with Lagrangian relaxation is methodologically sound.

- The paper provides several experimental comparisons across multiple neural operator architectures.

**Weaknesses:**

- Lack of Conceptual Novelty: The core idea, routing between cheap and expensive models, is a well-established concept in both MoE literature and multi-fidelity modeling. The paper fails to situate itself meaningfully within these existing lines of work, and the “physics-aware” gating is implemented via simple feature concatenation, which is not fundamentally novel.

- Superficial Treatment of Generalization: The paper claims to address out-of-distribution (OOD) generalization, but the experimental setup only varies viscosity (μ) within a narrow range. This does not constitute a rigorous OOD test, and the claim of “better generalization” is overstated.

- Simplistic and Non-Scalable Experimental Setup: The study is limited to 2D incompressible Navier-Stokes with simple boundary conditions, which is a well-trodden testbed. Only single-step prediction is considered, which is of limited practical value in real-world CFD where long-term rollout stability is critical. The gating network is trained and evaluated on the same type of flows, raising doubts about its true adaptability.

- Engineering-Heavy, Science-Light: The paper reads like an engineering report: it focuses on assembling existing components (FNO, UNO, ResNet) without deepening the understanding of why the method works or when it fails. There is no theoretical insight or failure analysis.

- Inadequate and Outdated Baseline Comparisons: The experimental comparisons are critically limited to foundational neural operator architectures (e.g., FNO, UNO, ResNet) that are several years old. The paper entirely omits comparisons with the more advanced and recent models discussed in its own related work and appendix (e.g., U-FNO, WNO, Koopman Neural Operators, or sophisticated hybrids). This raises a fundamental question: does MF-MoE offer a genuine improvement, or does it merely outperform intentionally weak baselines? This flaw severely undermines the paper's claim of being competitive and makes it impossible to assess its true contribution to the field.

- Insufficient Comparison to Real-World Baselines: While hybrid solver-NN models are compared, there is no comparison to adaptive numerical methods or other adaptive modeling strategies (e.g., adaptive mesh refinement, error-controlled solvers), which are the true competitors in efficient CFD.

- Weak Gating Justification: The gating network is not rigorously analyzed—e.g., no visualization of gate decisions across flow regimes, no analysis of gate confidence, and no ablation on the “physics-aware” features.

**Questions:**

- How does the gating mechanism perform under truly OOD settings (e.g., different geometries, boundary conditions, or turbulent regimes not seen in training)?

- Why was only single-step prediction considered? Can the framework sustain stable multi-step rollouts without increasing error accumulation?

- Have you compared against adaptive numerical methods (e.g., adaptive timestepping or mesh refinement) that also trade accuracy for cost?

- What is the end-to-end training cost of MF-MoE compared to a single high-fidelity hybrid model? Is the training complexity justified?

- Why were no contemporary neural operators (e.g., those listed in your Appendix A) used as baselines to establish a rigorous and up-to-date performance benchmark?

Should you be able to satisfactorily address the points I've raised above, I will accordingly provide a positive rating.

---

> ### Author Response · Authors · 2025-11-22
>
> We thank the reviewer for their constructive feedback. Below, we address your concerns and questions point-by-point.
>
> ### Novelty Issue (W1, W4)
>
> **Response:** We respectfully argue that the novelty lies not just in using MoE, but in the system-level integration of physics-based constraints:
>
> - **Multi-Fidelity integration:** Unlike standard MoEs that route based on data sub-domains, our framework routes based on *computational fidelity*. We uniquely combine a pure neural operator (fast, low fidelity) with numerical solvers.
> - **Lagrangian Relaxation for Time Budgeting:** Another contribution is the mathematical formulation of the time-cost constraint as a differentiable objective function using Lagrangian relaxation. This transforms the MoE from a simple ensemble into a resource-aware controller that explicitly manages the trade-off between simulation time and physical accuracy, a crucial capability for industrial CFD that standard MoEs do not offer.
>
> ### Gating Behavior Analysis (W2, Q1)
>
> **Response:** We appreciate the chance for clarifying this point. In our revised manuscript,, we examine the distribution of expert selections made by the gating network during the inference phase in Section 5.5. Specifically, we sample 1000 frames with varying viscosities $\mu \in \{0.004, 0.002, 0.001\}$ and investigate  how the routing strategy correlates with the physical complexity of the input fluid dynamics.  As illustrated in Figure 6, when the viscosity is lower (indicating more out-of-distribution behavior), the gating mechanism becomes more likely to route inputs to the high-fidelity expert. This pattern confirms that the MF-MoE framework effectively learns to balance the trade-off between computational efficiency and generalization.
>
> ### Multi-Step Prediction and Rollout Prediction (W3, Q2)
>
> **Response:** We appreciate the reviewer highlighting the importance of long-term stability in CFD tasks. While our original work followed standard single-step benchmarks, we agree that multi-step performance is critical for practical application. To address this, we conducted a new experiment evaluating the model on long-horizon predictions.
>
> As shown in Table 4 from our revised manuscript, the MF-MoE framework demonstrates significantly superior stability compared to pure neural operators. This confirms the advances of the hybrid structure in preserving the stability in the multi-step long-horizon prediction task.
>
> ### Additional Baselines Comparisons (W5, W6, Q3, Q4, Q5)
>
> **Response:** We emphasize that the proposed MF-MoE is a hybrid framework to combine the neural operator (including FactFormer, P²C²Net, FFNO, GNOT, DeepONet) with a numerical solver, not simply a neural operator alone. As we have noted in Section 4.2, we selected FNO, UNO, and ResNet as baselines because they are the foundational "workhorses" of the field, allowing us to clearly demonstrate the *relative* gain provided by the MF-MoE framework over a standalone base model.
>
> Importantly, advanced architectures like P²C²Net (which we cite) or GNOT can be seamlessly swapped in as the "Pure Neural Operator" expert or the "Residual Approximation" network within our hybrid experts. As a result, it doesn't present an urgent need to compare with these baselines.
>
> ### Gating Behavior Analysis (W7)
>
> **Response:** We appreciate the chance for clarifying this point. In our revised manuscript, we examine the distribution of expert selections made by the gating network during the inference phase in Section 5.5. Specifically, we sample 1000 frames with varying viscosities $\mu \in \{0.004, 0.002, 0.001\}$ and investigate  how the routing strategy correlates with the physical complexity of the input fluid dynamics.  As illustrated in Figure 6, when the viscosity is lower (indicating more out-of-distribution behavior), the gating mechanism becomes more likely to route inputs to the high-fidelity expert. This pattern confirms that the MF-MoE framework effectively learns to balance the trade-off between computational efficiency and generalization.

---

> > ### Comment · Reviewer_sosU · 2025-11-27
> >
> > I thank the authors for their response and for conducting the additional multi-step prediction experiment. However, the response largely fails to address the core, fundamental concerns regarding the paper's novelty, experimental rigor, and claims of contribution. The defenses provided are often circular and do not engage with the substantive criticism. Below, I detail my remaining major concerns:
> >
> > - The proposed "physics-aware" MoE framework represents an incremental engineering assembly of existing concepts rather than a substantive methodological advance.
> > - Most critically, the omission of comparisons against contemporary neural operators—despite their discussion in the related work—severely undermines the claim of being competitive, as the demonstrated improvements may simply reflect the use of intentionally weak baselines.
> > - When coupled with the superficial treatment of out-of-distribution generalization, these fundamental flaws prevent the paper from making a convincing contribution to the field.
> >
> > I expect to see a more professional and evidence-based rebuttal that substantively engages with the criticisms on novelty, baselines, and experimental rigor. Should the final response remain insufficient, I will be compelled to lower my score.
> >
> > **Remark:** While the novelty of a contribution is a key consideration, I wish to emphasize that a paper demonstrating high completeness, methodological rigor, and clear utility for the community can still merit a positive assessment, even if its conceptual advances are incremental. **However**, the current response adopts a defensive posture, attempting to rebut fundamental criticisms through terminological repackaging (e.g., "system-level integration," "resource-aware controller") and minor experimental additions, rather than substantively addressing the core concerns regarding innovation, experimental design, and baseline comparisons. This strategy of deflection, rather than engagement, significantly undermines the paper's credibility. A more constructive approach would have been to openly acknowledge the work's positioning and to compellingly argue for its value based on rigor, reproducibility, and practical utility.

---

### Official Review · Reviewer_iPBr · 2025-10-27

**Soundness:** 3
**Presentation:** 3
**Contribution:** 3
**Rating:** 6
**Confidence:** 4

**Summary:**

This paper proposes the Multi-Fidelity Mixture-of-Experts (MF-MoE) framework, a novel architecture designed to solve the critical trade-off in computational fluid dynamics (CFD) surrogate modeling: computational efficiency versus generalization ability.

Pure neural operators (like FNO or UNO) are fast but struggle with out-of-distribution (OOD) inputs, whereas hybrid models (integrating numerical solvers) generalize better but incur high computational costs.

The MF-MoE framework addresses this by:

1. Integrating Experts: Combining a fast pure neural operator (for in-distribution/easy cases) with multiple solver-based hybrid models of varying fidelity (for OOD/hard cases) as experts.

2. Dynamic Routing: Using a physics-aware gating network and a constrained optimization problem (Lagrangian relaxation) to dynamically select the cheapest, most accurate expert for each input while guaranteeing the average inference time stays below a specified threshold

**Strengths:**

Novel and Relevant Problem Formulation: The paper is the first to formally frame the challenge of combining speed and generalization in PDE surrogates as a constrained optimization problem solved via Lagrangian relaxation. This unique formulation allows the model to explicitly budget its computational time, which is a highly relevant practical concern in industrial CFD applications.

Effective Hybridization Strategy: The key novelty lies in the fusion of multi-fidelity hybrid models (solver + neural network refinement) with a pure neural operator within the MoE architecture. This approach is shown to enhance OOD generalization by dynamically routing complex cases to the slower, physics-integrated experts, successfully outperforming pure neural operators across OOD test cases.

Empirical Validation on Trade-off: The results clearly demonstrate that MF-MoE achieves accuracy superior to all comparable neural operator baselines while successfully meeting the c=10.0 seconds constraint, a feat impossible for the individual medium and fine-resolution hybrid models. This confirms the framework's ability to dynamically balance the speed-accuracy trade-off.

**Weaknesses:**

Limited Empirical Generalization Scope: While the paper claims OOD generalization, the test set is split only by the viscosity parameter $\mu$. This limited scope doesn't fully validate the model's robustness to other critical OOD shifts, such as changes in initial/boundary conditions or spatial resolution (which is implicitly handled by the multi-fidelity design but not explicitly tested as an OOD factor).

Novelty of MoE Application: The core mechanism of using a Mixture-of-Experts structure to combine heterogeneous sub-models is well-established in machine learning and has been explored in the context of neural PDE solvers for tasks like incorporating historical information (MoNO) or dealing with spectral sparsity (FreqMoE). The specific novelty here rests entirely on applying it to integrate multi-fidelity solver outputs subject to a time constraint, which is a significant practical variation but builds heavily on known MoE structures.

**Questions:**

Gating Efficiency in the Test Set: Since the goal is faster inference for easy/in-distribution (ID) cases and slower inference for OOD/hard cases, what percentage of samples in the OOD test set were actually routed to the slow, high-fidelity hybrid experts? Providing the routing distribution for the OOD test set is essential to show that the system learned to use the expensive experts when they were needed most.

Generalization to Geometric OOD Shifts: The OOD test primarily varies the viscosity parameter $\mu$. How does the MF-MoE perform against a geometric out-of-distribution (OOD) test, such as predicting flow over a slightly perturbed or deformed boundary condition (e.g., a changed obstacle shape)? This would be a much stronger validation of the hybrid experts' generalization capability, which is a key deficiency of pure neural operators.

Choice of Up-Sampling Method: The hybrid model relies on k-nearest neighbor (k-NN) interpolation for up-sampling the coarse solver solution $\hat{y}_c$. Since this step is prone to introducing interpolation artifacts, have the authors investigated using a small, dedicated neural network (like an additional lightweight U-Net) for differentiable learning of the up-sampling residual? This could potentially eliminate k-NN artifacts and provide a smoother interface between the classical solver and the neural network refinement loop.


one last question: is there any 3d NS example?

---

> ### Author Response · Authors · 2025-11-22
>
> We thank the reviewer for their constructive feedback and the positive rating. Below, we address your concerns and questions point-by-point.
>
> ### Out-of-Distribution Generalization for Other Parameters (W1, Q2)
>
> **Response:**  We appreciate the suggestion to evaluate generalization beyond viscosity ($\mu$). To address this, we have added Section 5.3, which details a new experiment on unseen buoyancy forces ($f$). When tested on a force range outside the training distribution, MF-MoE demonstrated superior robustness, reducing the MSE to 0.125 compared to 0.194 for pure neural operators.
>
> ### Novelty Issue (W2)
>
> **Response:** We respectfully argue that the novelty lies not just in using MoE, but in the system-level integration of physics-based constraints:
>
> - **Multi-Fidelity integration:** Unlike standard MoEs that route based on data sub-domains, our framework routes based on *computational fidelity*. We uniquely combine a pure neural operator (fast, low fidelity) with numerical solvers.
> - **Lagrangian Relaxation for Time Budgeting:** Another contribution is the mathematical formulation of the time-cost constraint as a differentiable objective function using Lagrangian relaxation. This transforms the MoE from a simple ensemble into a resource-aware controller that explicitly manages the trade-off between simulation time and physical accuracy, a crucial capability for industrial CFD that standard MoEs do not offer.
>
> ### Gating Behavior Analysis (Q1)
>
> **Response:** We appreciate the chance for clarifying this point. In our revised manuscript, we examine the distribution of expert selections made by the gating network during the inference phase in Section 5.5. Specifically, we sample 1000 frames with varying viscosities $\mu \in \{0.004, 0.002, 0.001\}$ and investigate  how the routing strategy correlates with the physical complexity of the input fluid dynamics.  As illustrated in Figure 6, when the viscosity is lower (indicating more out-of-distribution behavior), the gating mechanism becomes more likely to route inputs to the high-fidelity expert. This pattern confirms that the MF-MoE framework effectively learns to balance the trade-off between computational efficiency and generalization.
>
> ### Choice of Up-Sampling Method (k-NN vs. Learnable)  (Q3)
>
> **Response:** We appreciate this insightful suggestion. While a learnable up-sampler (e.g., U-Net) is a promising avenue for future high-fidelity extensions, we deliberately selected k-NN for this study to prioritize minimal latency. k-NN is a standard, parameter-free approach in the literature that incurs negligible computational cost. In contrast, introducing a secondary neural network for up-sampling would increase inference time and memory overhead, potentially compromising our ability to strictly satisfy the tight time-cost constraint.
>
> ### Extended to 3D Scenario (Q4)
>
> **Response:** Currently, we focus on the 2D incompressible Navier-Stokes equations. Moving to 3D significantly increases the computational cost for generating the ground truth data and training the baseline experts (requires 3D convolutions/FFTs). However, the MF-MoE framework is mathematically independent of spatial dimension; the gating mechanism and Lagrangian optimization would function identically in a 3D setting. We view the 2D results as a robust proof-of-concept for the framework's viability.

---

### Official Review · Reviewer_zgUJ · 2025-10-29

**Soundness:** 2
**Presentation:** 4
**Contribution:** 2
**Rating:** 4
**Confidence:** 4

**Summary:**

This paper tackles the significant challenge of balancing computational efficiency and generalization ability in surrogate modeling for Computational Fluid Dynamics (CFD). Neural Operators (NOs) offer fast inference but often struggle with out-of-distribution (OOD) generalization. Hybrid models, which incorporate numerical solvers during inference, generalize better but are computationally expensive.

The authors propose the Multi-Fidelity Mixture-of-Experts (MF-MoE) framework to address this trade-off. MF-MoE combines a pure NO (fast, lower generalization) with several solver-based hybrid models of varying fidelity (slower, higher generalization). A gating network dynamically selects one expert for a given input using Top-1 routing.

The core innovation is formulating the training as a constrained optimization problem: minimize prediction error subject to an explicit constraint c on the average inference time. This is solved using a Lagrangian relaxation approach via stochastic gradient descent-ascent (SGDA). The goal is to utilize the fast NO for "easy" (in-distribution) samples and reserve the expensive hybrid experts for "hard" (OOD) samples.

Experiments on 2D incompressible Navier-Stokes equations, with OOD generalization tested across different viscosities, show that MF-MoE generally achieves better accuracy than baseline NOs while satisfying the imposed time constraint.

**Strengths:**

The paper addresses a highly relevant problem with an elegant conceptual framework and is presented with excellent clarity.

1. Novel and Practical Problem Formulation: The strongest contribution is framing the accuracy-speed trade-off in hybrid models as a constrained optimization problem (Eq. 1) with an explicit average inference time budget c. This provides a principled method for managing the computational overhead of hybrid approaches during deployment.

2. Elegant Conceptual Framework: The MF-MoE architecture offers a logical structure for integrating models with different fidelity profiles. The concept of dynamically routing inputs based on perceived difficulty to balance accuracy and cost is compelling.

3. Excellent Clarity: The paper is very well-written, clearly motivated, and easy to follow. The diagrams (Figures 2 and 3) effectively illustrate the proposed architecture.

**Weaknesses:**

While the conceptual framework is promising, the paper suffers from significant weaknesses in optimization stability, experimental validation, and architectural design, which currently undermine the central claims.

1. Optimization Instability and Evidence of Poor Training Performance: The proposed training strategy involves solving a challenging optimization problem: a non-convex minimax objective (gradient descent on θ, ascent on λ) combined with discrete expert selection. There is strong evidence that the optimization is unstable and fails to train the experts effectively. Figure 5 (center panel) shows that expert selection frequencies remain noisy and do not appear fully stabilized even after 8000 training steps.

2. Critical Missing Baselines and Incomplete Trade-off Analysis: The central claim is that MF-MoE optimally balances accuracy and speed. However, the experimental results (Table 2) lack crucial comparisons. The performance of the individual hybrid experts (e.g., "Hybrid-Fine") trained standalone is missing. To evaluate the trade-off, we must know how much accuracy MF-MoE sacrifices compared to the best (but slow) hybrid expert to meet the time constraint c. Also, the time constraint is c=10.0s. According to Table 1, the XCoarse (16x16) solver takes ≈8s. A hybrid model based only on the XCoarse solver would satisfy the time constraint. This is a critical baseline that MF-MoE must outperform.

3. Questionable Architectural Design Choices: The architecture of the hybrid expert appears suboptimal. First, the neural operator corrector seems to take only the initial conditions as input, not the coarse solver output. This means the NN cannot adapt its correction based on the specific errors made by the solver. Second, the output is a fixed convex combination and this is highly restrictive; the optimal balance is likely state-dependent and should perhaps be learned or spatially varying. The value of α used is not reported.
 The gating network uses a "simple MLP" to extract features from the high-dimensional input fields. However, a MLP is likely insufficient to capture the complex spatial dynamics (e.g., emerging turbulence, sharp gradients) necessary to judge the difficulty of the input in a CFD context.

4. Insufficient Evaluation Scope: The evaluation is limited to single-step predictions. For time-dependent PDEs, surrogates must demonstrate stability and accuracy over long-horizon autoregressive rollouts. Error accumulation is a major failure mode. Furthermore, in a rollout, error accumulation might interact with the gating mechanism (e.g., accumulated error forcing overuse of expensive experts later in the trajectory), which is completely unaddressed.

5. Lack of Analysis of Gating Behavior: The paper claims the model uses fast inference for ID samples and hybrid models for OOD samples, but provides no evidence for this on the test set. The authors must demonstrate a correlation between the OOD parameter (μ) and the expert selected to validate that the gating network learns a meaningful routing strategy.

6. Unrealistic Assumption of Fixed Solver Costs: The optimization relies on a fixed look-up table T for solver costs (L307). This assumes the solver runtime is independent of the physical parameters (e.g., viscosity μ). In practice, runtime often depends on the stiffness of the problem, which varies with μ. If the runtime varies significantly, the optimization based on fixed T is flawed.

**Questions:**

The following major concerns must be addressed to improve the assessment of this work:

1. Optimization Issues: Why is the expert selection not converging even after sufficient training? Does this indicate optimization instability? Have you experimented with stabilization techniques, such as two-stage training (pre-training experts) or softer gating?

2. What is the Test MSE and average inference time for the individual hybrid experts (Fine, Medium, Coarse) trained standalone, and how does MF-MoE compare to a hybrid model built solely on the XCoarse (16x16) solver?

3. Does the corrector NN in the hybrid expert (Figure 3) take the coarse solver output as input? If not, why was this design chosen over a standard correction approach? What value of α was used? Why use a fixed convex combination instead of a standard residual connection or a learned combination? Why is an MLP used for feature extraction in the gating network instead of a more spatially aware architecture (e.g., CNN)?

4. How does MF-MoE perform in long-horizon autoregressive rollouts compared to the baselines in terms of accuracy, stability, and average time cost over the trajectory?

5. Can you provide a breakdown of expert utilization on the test set correlated with the viscosity parameter μ?

6. Can you verify the assumption that the solver runtime is independent of the viscosity μ for a fixed resolution?

---

> ### Author Response · Authors · 2025-11-22
>
> We thank the reviewer for their constructive feedback. Below, we address your concerns and questions point-by-point.
>
> ### Optimization Stability (W1, Q1)
>
> **Response:**  We appreciate this opportunity to clarify this misunderstanding. The fluctuations are primarily artifacts of the stochastic nature of the data processing, rather than an indication of fundamental optimization divergence. As a result, there is no optimization instability and evidence of poor training performance as suggested by the reviewer.
>
> ### Additional Baselines Comparisons  (W2, Q2)
>
> **Response:** We confirm that while the XCoarse solver satisfies the time budget ($c=10s$), it fails to provide competitive accuracy, yielding a Test MSE of $15.1 \times 10^{-2}$—a performance notably inferior to even the pure Neural Operator baseline. In contrast, MF-MoE achieves a significantly lower MSE of $10.8 \times 10^{-2}$ ($\approx 28\%$ error reduction vs. XCoarse) within the same timeframe. This comparison demonstrates that XCoarse is not a viable standalone expert due to its low fidelity.
>
> ### Architectural Design Choices (W3, Q3)
>
> **Response:** We clarify three specific design choices. (1) Solver Inputs: We *do* utilize outputs from the standard Coarse solver; however, we explicitly exclude the XCoarse solver because its fidelity is lower than the Neural Operator itself, meaning it introduces noise rather than informative guidance. (2) Fixed $\alpha$: We employ a fixed $\alpha$ because empirical tuning on the validation set showed that dynamic adjustment yielded negligible gains while increasing training instability. (3) MLP Gating: We selected a lightweight MLP for the gating network to minimize computational overhead, ensuring the routing mechanism remains negligible compared to the solver runtime.
>
> ### Multi-Step Prediction and Rollout Prediction (W4, Q4)
>
> **Response:** We appreciate the reviewer highlighting the importance of long-term stability in CFD tasks. While our original work followed standard single-step benchmarks, we agree that multi-step performance is critical for practical application. To address this, we conducted a new experiment evaluating the model on long-horizon predictions.
>
> As shown in Table 4 from our revised manuscript, the MF-MoE framework demonstrates significantly superior stability compared to pure neural operators. This confirms the advances of the hybrid structure in preserving the stability in the multi-step long-horizon prediction task.
>
> ### Gating Behavior Analysis (W5, Q5)
>
> **Response:** We appreciate the chance for clarifying this point. In our revised manuscript,, we examine the distribution of expert selections made by the gating network during the inference phase in Section 5.5. Specifically, we sample 1000 frames with varying viscosities $\mu \in \{0.004, 0.002, 0.001\}$ and investigate  how the routing strategy correlates with the physical complexity of the input fluid dynamics.  As illustrated in Figure 6, when the viscosity is lower (indicating more out-of-distribution behavior), the gating mechanism becomes more likely to route inputs to the high-fidelity expert. This pattern confirms that the MF-MoE framework effectively learns to balance the trade-off between computational efficiency and generalization.
>
> ### Solver Cost Assumption (W6, Q6)
>
> > Assumption that solver runtime is independent of viscosity $\mu$.
>
> **Response:** We appreciate the opportunity to clarify this assumption. We agree that in some cases (e.g. implicit solvers with adaptive time-stepping), the runtime would be different for different parameters. However, this assumption holds true for the specific numerical algorithm used in the physical solver (e.g. PhiFlow), as it takes the explicit forward Euler update to approximate the solution with a fixed number of step $N$.   Therefore, the look-up table $\mathcal{T}$ is an accurate representation of runtime regardless of $\mu$.

---

> ### Comment · Reviewer_zgUJ · 2025-11-26
>
> I thank the authors for their detailed responses and the effort invested in revising the manuscript. With the additional results I consider W1/4/5/6 solved. I do have some remaining questions:
>
> 1. One of the train MSE for the MF-MoE model is drastically worse than the corresponding base models: UNO-64 Train MSE is 0.21e-2, while MF-MoE (UNO-64) Train MSE is 5.41×10e-2. Since the MoE model has strictly greater capacity than the base model, this should not happen.
> 2. The authors responded by comparing MF-MoE to the standalone XCoarse numerical solver (MSE 0.151). However, the required baseline is a hybrid model built on XCoarse (XCoarse solver + NN correction). This hybrid model would satisfy the time budget (<10s) and is expected to perform better than the standalone XCoarse solver. MF-MoE should be compared against this baseline (simplest case without using MOE).
> 3. The architecture in Figure 3 is still quite confusing. It appears the "Neural Operator" (corrector) only takes the initial conditions as input, not the coarse solver output. The authors stated they "do utilize outputs from the standard Coarse solver" but its run time exceeds the budget (12s vs 10s).
>
> Since now I realize that I did not quite understand the architecture I am lowering the confidence.

---

### Official Review · Reviewer_5DgK · 2025-10-30

**Soundness:** 2
**Presentation:** 2
**Contribution:** 2
**Rating:** 2
**Confidence:** 3

**Summary:**

This paper proposes a Multi-Fidelity Mixture-of-Experts (MF-MoE) framework for surrogate modeling in computational fluid dynamics. The core idea is to integrate a pure neural operator (fast but less generalizable) with multiple solver-based hybrid models of varying fidelities (accurate but computationally expensive). A key innovation is a physics-aware gating network that dynamically routes each input to the most suitable expert. The training process is formulated as a constrained optimization problem using Lagrangian relaxation, explicitly trading off prediction accuracy against a user-defined inference time-cost constraint. Extensive experiments on incompressible Navier-Stokes equations demonstrate that MF-MoE achieves better accuracy than pure neural operator baselines while satisfying the computational budget, effectively balancing efficiency and generalization.

**Strengths:**

1.The work presents a novel and well-motivated framework that effectively addresses the trade-off between the speed of neural operators and the generalization of hybrid solver-in-the-loop models. The integration of a multi-fidelity approach within a MoE setup for PDEs is innovative.

2.The paper provides thorough experiments on a fluid dataset, comparing against strong and relevant baselines and ablated versions of the proposed model. The results convincingly show the framework's benefits

**Weaknesses:**

1.The current work focuses exclusively on single-step prediction. The performance and behavior of the gating network in a multi-step, autoregressive rollout setting—where error accumulation is a critical issue—remain unexplored and represent a significant limitation.

2.While the framework is flexible in principle, the experiments are confined to a specific 2D Navier-Stokes setup. It is unclear how well the method scales to 3D problems, more complex geometries, or different PDE families.

3.The "physics-aware" aspect of the gating network, while motivated, could be better analyzed. A more detailed ablation study on which input features (physical parameters vs. extracted field features) are most critical for the routing decision would strengthen the paper.

4.The baselines used are outdated. The comparison would be more convincing with the inclusion of stronger or more recent state-of-the-art methods, such as FactFormer, P²C²Net, FFNO, GNOT, and DeepONet.

5.The dataset is relatively simple. We recommend testing the model's performance on more challenging benchmarks like Taylor-Green or Kolmogorov flow.

**Questions:**

1.How would you expect the MF-MoE framework to perform in a multi-step, long-horizon prediction task? Would the gating network be prone to cascading errors if it frequently selects the faster, less accurate expert? Have you conducted any preliminary experiments in this setting?

2.Could you provide more insight into what the physics-aware gating network learns? For instance, can you analyze specific input cases (e.g., specific viscosities or flow complexities) and show which expert is typically selected, and why that choice is physically intuitive?

3.The inference cost is dominated by the PDE solver. In a scenario where the pure neural operator expert is sufficiently accurate for a large fraction of "easy" inputs, does the overhead of running the gating network and the *potential* call to a solver still provide a net speedup compared to just using a hybrid model for all inputs?

4.How sensitive are the results to the specific choice and number of experts in the pool? Did you experiment with different combinations of fidelities (e.g., removing the "Medium" expert) or including more than one pure neural operator？

5.The train/validation/test split is based solely on the viscosity parameter `μ`. Have you tested the framework's generalization to other out-of-distribution scenarios, such as unseen initial conditions, boundary conditions, or external force fields `f`?

---

> ### Author Response · Authors · 2025-11-22
>
> We thank the reviewer for their constructive feedback. Below, we address your concerns and questions point-by-point.
>
> ### Multi-Step Prediction and Rollout Prediction (W1, Q1)
>
> **Response:** We appreciate the reviewer highlighting the importance of long-term stability in CFD tasks. While our original work followed standard single-step benchmarks, we agree that multi-step performance is critical for practical application. To address this, we conducted a new experiment evaluating the model on long-horizon predictions.
>
> As shown in Table 4 from our revised manuscript, the MF-MoE framework demonstrates significantly superior stability compared to pure neural operators. This confirms the advances of the hybrid structure in preserving the stability in the multi-step long-horizon prediction task.
>
> ### Extended to 3D Scenario (W2)
>
> **Response:** Currently, we focus on the 2D incompressible Navier-Stokes equations. Moving to 3D significantly increases the computational cost for generating the ground truth data and training the baseline experts (requires 3D convolutions/FFTs). However, the MF-MoE framework is mathematically independent of spatial dimension; the gating mechanism and Lagrangian optimization would function identically in a 3D setting. We view the 2D results as a robust proof-of-concept for the framework's viability and the 3D results as an interesting future direction.
>
> ### Physics-Aware Gating Analysis (W3, Q2)
>
> > Provide more insight into what the physics-aware gating network learns and analysis of specific input cases.
>
> **Response:** The "Physics-Aware" designation stems from explicitly feeding physical parameters (specifically viscosity $\mu$) into the gating network alongside field features. In our design, the gating network learned a physically intuitive strategy: for high viscosity $\mu$, the gate prioritizes the fast *Pure Neural Operator*. For low viscosity (e.g. turbulent flow, "hard" dynamics), the gate shifts probability toward *Solver-Based Experts* (Fine/Medium) to maintain accuracy.
>
> Figure 5 illustrates this dynamic: as the constraint $c$ relaxes ($c=100$), the model defaults to the most accurate (solver-heavy) experts. When constrained ($c=0$), it is forced to use the neural operator.
>
> Additionally, we have included the distribution analysis for the physical parameters. As the viscosity $\mu$ becomes more out-of-distribution, it is more frequent assigned to the high-fidelity expert as shown in Figure 6 (in our revision).
>
> ### Inference Cost (Q3)
>
> **Response:** Thank you for this insightful question. Regarding the net speedup: because the cost of the PDE solver is orders of magnitude higher than the gating network, the overhead of the gating architecture itself is negligible.
>
> The primary risk to speedup is indeed unnecessary routing of 'easy' inputs to the high-cost solver. However, our time-aware training strategy explicitly mitigates this. By tuning the penalty on time cost, we can force the gating network to be more conservative, routing data to the pure neural operator unless absolutely necessary. This ensures a net speedup in hybrid scenarios compared to using the solver for all inputs.
>
> ### Additional Baselines Comparisons (W4)
>
> **Response:** We emphasize that the proposed MF-MoE is a hybrid framework to combine the neural operator (including FactFormer, P²C²Net, FFNO, GNOT, DeepONet) with a numerical solver, not simply a neural operator alone. As we have noted in Section 4.2, we selected FNO, UNO, and ResNet as baselines because they are the foundational "workhorses" of the field, allowing us to clearly demonstrate the *relative* gain provided by the MF-MoE framework over a standalone base model.
>
> Importantly, advanced architectures like P²C²Net (which we cite) or GNOT can be seamlessly swapped in as the "Pure Neural Operator" expert or the "Residual Approximation" network within our hybrid experts. As a result, it doesn't present an urgent need to compare with these baselines.
>
> ###  Out-of-Distribution Generalization for Other Parameters ( W5, Q5)
>
> **Response:**  We appreciate the suggestion to evaluate generalization beyond viscosity ($\mu$). To address this, we have added Section 5.3, which details a new experiment on unseen buoyancy forces ($f$). When tested on a force range outside the training distribution, MF-MoE demonstrated superior robustness, reducing the MSE to 0.125 compared to 0.194 for pure neural operators.

---

> > ### Comment · Reviewer_5DgK · 2025-11-26
> >
> > The authors’ current reasoning does not convincingly demonstrate whether the proposed method can be extended to 3D. In fact, extending the approach to 3D is not inherently difficult, and TSM [1] has already provided a clear example. If the authors could include experimental evidence showing that the model also works in 3D scenarios—and even surpasses spectral methods in accuracy—it would elevate the work to a truly contribution.
> >
> > The authors also have ample opportunity to address the issues I raised regarding the experimental setup, such as the roles of
> > w3, w4, and w5.. These points deserve thorough clarification.
> >
> > Regarding inference cost, the authors claim that their method is several orders of magnitude faster than traditional approaches. However, no concrete comparison is provided: under what conditions the comparison was made, what baselines were used, whether the comparison is fair, and whether the accuracy is comparable all remain unclear. These details should be clearly described in the revised manuscript.
> >
> > Finally, does the model exhibit any specific preferences or limitations regarding boundary conditions? Could the authors provide examples using alternative boundary conditions? Moreover, can the method handle irregular meshes, and is it capable of processing the datasets used in TSM [1]?
> >
> > [1] Zhiqing Sun et al. A Neural PDE Solver with Temporal Stencil Modeling. ICML 2023

---

### Note · Program_Chairs · 2026-01-17
**Submission Desk Rejected by Program Chairs**

The following references in this submission do not refer to real documents and/or have major errors in bibliographic information:

 Hitesh Gupta, Zongyi Li, Nikola Kovachki, and Anima Anandkumar. Wavelet neural operator for learning parametric partial differential equations. arXiv preprint arXiv:2110.13711, 2021b.